# Disruption of the Mammalian Ccr4–Not Complex Contributes to Transcription-Mediated Genome Instability

**DOI:** 10.3390/cells12141868

**Published:** 2023-07-17

**Authors:** Nafiseh Chalabi Hagkarim, Morteza Chalabi Hajkarim, Toru Suzuki, Toshinobu Fujiwara, G. Sebastiaan Winkler, Grant S. Stewart, Roger J. Grand

**Affiliations:** 1Institute for Cancer and Genomic Sciences, The Medical School, University of Birmingham, Birmingham B15 2TT, UK; g.s.stewart@bham.ac.uk; 2Department of Medicine Haematology & Oncology, Columbia University, New York City, NY 10032, USA; mc5196@cumc.columbia.edu; 3Division of RNA and Gene Regulation, Institute of Medical Science, The University of Tokyo, Tokyo 108-8639, Japan; torusuzuki@g.ecc.u-tokyo.ac.jp; 4Laboratory of Biochemistry, Kindai University, Higashi-Osaka City 577-8502, Japan; tosinobu@phar.kindai.ac.jp; 5School of Pharmacy, University of Nottingham, Nottingham NG7 3RD, UK; sebastiaan.winkler@nottingham.ac.uk

**Keywords:** CNOT complex, CNOT1, CNOT7, CNOT8, genome instability, DNA repair, transcription

## Abstract

The mammalian Ccr4–Not complex, carbon catabolite repression 4 (Ccr4)-negative on TATA-less (Not), is a large, highly conserved, multifunctional assembly of proteins that acts at different cellular levels to regulate gene expression. It is involved in the control of the cell cycle, chromatin modification, activation and inhibition of transcription initiation, control of transcription elongation, RNA export, and nuclear RNA surveillance; the Ccr4–Not complex also plays a central role in the regulation of mRNA decay. Growing evidence suggests that gene transcription has a vital role in shaping the landscape of genome replication and is also a potent source of replication stress and genome instability. Here, we have examined the effects of the inactivation of the Ccr4–Not complex, via the depletion of the scaffold subunit CNOT1, on DNA replication and genome integrity in mammalian cells. In CNOT1-depleted cells, the elevated expression of the general transcription factor TATA-box binding protein (TBP) leads to increased RNA synthesis, which, together with R-loop accumulation, results in replication fork slowing, DNA damage, and senescence. Furthermore, we have shown that the stability of TBP mRNA increases in the absence of CNOT1, which may explain its elevated protein expression in CNOT1-depleted cells. Finally, we have shown the activation of mitogen-activated protein kinase signalling as evidenced by ERK1/2 phosphorylation in the absence of CNOT1, which may be responsible for the observed cell cycle arrest at the border of G1/S.

## 1. Introduction

The yeast Ccr4–Not complex is a large (1.9-MDa), highly conserved multifunctional assembly of proteins involved in many different aspects of mRNA metabolism. These include the repression and activation of transcription initiation, and the control of mRNA elongation and deadenylation-dependent mRNA turnover; it also possesses ubiquitin-protein transferase activity [1,2,3,4,5,6]. Most of the original studies conducted within this field have concentrated on the role of the Ccr4–Not complex in *Saccharomyces cerevisiae*. In yeast, the complex consists of core subunits, comprising Ccr4 (carbon catabolite repression), Caf proteins (Ccr4-associated factor) (Caf1, Caf40, Caf130) and Not proteins (Not1, Not2, Not3, Not4, and Not5), as well as several less strongly associated components which are probably interacting partners; reviewed [7,8]. Not1 is the largest subunit of the complex (>200 kD) and forms a scaffold for assembly of the other components [9]. The Caf1 and Ccr4 proteins have deadenylase activity, whilst the Not4 protein is the E3 ligase [10,11,12,13,14,15]. No clear function has been assigned to the Not module comprising Not2, Not3 and Not5, although it does act as a cofactor for the deadenylation activity and may be involved in interaction with the ribosome [16,17]. The mammalian Ccr4–Not complex is broadly similar to that in yeast, except for the presence of two additional components(CNOT10 and CNOT11) which have no yeast equivalents. The mammalian E3 ligase is CNOT4. CNOT6/6L and CNOT7/8 have deadenylase activity. Yeast Caf130 appears to be unique and have no mammalian equivalent. Genetic approaches in yeast have demonstrated that the Not1–4 genes can repress global RNA polymerase II activity, such that their mutation or deletion increases the basal expression of many genes [18] The Ccr4–Not complex regulates transcript buffering in yeast, controlling gene expression by altering both the transcription elongation rate and the rate of mRNA decay [19,20,21,22,23]). Importantly, the Ccr4–Not complex can both negatively and positively regulate global gene expression by affecting both transcription initiation and elongation activities, with the Ccr4–Not complex regulating both RNA Pol I and RNA Pol II transcriptional activities [3,24,25,26].

The Ccr4–Not subunits associate with transcription regulatory factors (on the 5′ regions of various genes), such as TFIID, the SAGA (Spt–Ada–Gcn5 acetyltransferase) histone acetyltransferase complex, TATA binding protein-associated factor (TAF), TATA-binding protein (TBP), SRB/Mediator complex and PAF (RNA polymerase II-associated factor) complexes, reflecting the roles of the Ccr4–Not complex in transcription initiation as well as in elongation [27,28,29,30,31,32,33].The Ccr4–Not complex functionally and physically associates with TBP and TBP-associated components [29]. For example, TBP canonical binding is highly suppressed by Not1 in yeast [34,35]. Not1 has the ability to bind TBP in the TFIID and SAGA regulatory transcription factor complexes, which function at approximately 90% and 10% of the gene promoters, respectively [36,37,38]. TFIID is typically found on promoters of constitutive TATA-less housekeeping genes that are required for the maintenance of basic cellular functions, while SAGA-dominated genes are largely stress induced and TATA-containing.

Transcription may affect DNA replication if co-transcriptional products such as R-loops (three-stranded DNA–RNA hybrid structures) are not efficiently removed behind the transcription machinery. The collision of replication and transcription complexes with R-loops threatens the genome integrity by increasing the rate of mutagenesis, particularly over highly active promoters, such as those that occur in ribosomal genes (rDNA) [39,40]. Genome stability is normally protected against R-loop formation via sets of surveillance factors. For example, the Ccr4–Not complex functionally co-operates with the elongation factor TFIIS to rescue arrested RNA Pol II. TFIIS directly induces the resumption of elongation and helps RNA Pol II to pass through trapping sites by increasing its intrinsic nuclease activity and displacing transcripts [3,22,41,42,43,44,45].The interactions between the Ccr4–Not complex and R-Loop-processing factors, hnRNPs and the THO/TREX complex, indicate a potential role for the Ccr4–Not complex in the repair of R-loop structures [8,24,46,47]. Transcription-coupled repair (TC-NER) nucleases XPG and XPF can excise R-loops that block transcription, leaving single-strand DNA (ssDNA) gaps which can collapse into DNA double-strand breaks (DSBs) [48]. TC-NER is impaired by mutations in the Ccr4–Not complex in yeast, suggesting additional functions in DNA repair during transcription elongation [49]. Ccr4–Not complex mutants have been identified, which impair transcription elongation through long GC-rich genes with different gene length accumulation of *m*RNA (GLAM) ratios [49]. These long-GC-rich regions impede transcription elongation by forming R-loops, suggesting that the Ccr4–Not complex may promote elongation through GC-rich regions by preventing R-loop formation during transcription (Reese 2013).

The CNOT complex has also been shown to be more directly involved in the response to DNA damage. For example, in a screening of deletion mutants sensitizing *S. pombe* to DNA damage Caf1, Ccr4 and Caf40 were identified [50]. Similarly, Ccr4 interconnects genetically and/or physically with several yeast repair genes in an ionizing radiation-mediated damage response network [51]. Ccr4 acts in the same pathway as Chk1 in the response to hydroxy-urea-induced DNA replication stress. In mammalian cells, the depletion of CNOT6 leads to the resistance to cisplatin-induced apoptosis following Chk2 phosphorylation (on T68), whilst CNOT6 overexpression reduces Chk2 phosphorylation and increases the sensitivity of cells to DNA damage [52]. Tankyrase 1 Binding Protein 1 (TNKS1BP1 also known as Tab182) is a closely associated or integral component of the mammalian, but not yeast, complex. It contributes to IR-induced double-strand break repair by facilitating the binding of PARP1 to DNA-PKcs [53]. This protein has also been identified as a highly phosphorylated substrate by ATM/ATR after IR [54].

To better understand the role of the CNOT complex in the response to replication stress and the maintenance of genome stability in mammalian cells, we have taken a global approach. The depletion of the CNOT1 scaffold protein results in the destabilization of other proteins in the CCR4–Not complex [55]. We have now shown that, in the absence of CNOT1, there is an induction of R loops, leading to genome instability. In addition, the ablation of CNOT1 has direct effects on DNA replication, resulting in decreased fork speed in HeLa and MCF7 cells; this appears to be associated with increased transcription activity as treatment with transcription inhibitors, DRB and PHA, increases replication fork speed. R-loop accumulation in the absence of the CNOT1 leads to DNA double-strand breaks (DSBs) and the activation of the ATM/Chk2 signalling pathway. We have also shown that the depletion of CNOT1 leads to the cell cycle arrest at the border of G1/S where the DNA fork origin firing takes place with no capacity for elongation. At later times, CNOT1-depleted cells become senescent following the upregulation of p53, p21 and p16. Furthermore, we have shown that the loss of deadenylase components of the complex (CNOT7 and CNOT8) induces genome instability, as evidenced by the formation of micronuclei and DNA repair foci.

## 2. Methods and Materials

### 2.1. Cell Culture

HeLa S3 (ATCC-CCL-2.2 ™), HEK293T CNOT7-KO, HEK293T CNOT8-KO, and control HEK293T cells were grown in Dulbecco’s Modified Eagle’s Medium-high glucose (Sigmaaldrich, St. Louis, MI, USA) containing 8% foetal calf serum (FCS) (Sigmaaldrich) in 5% CO_2_ at 37 °C. For 70–80% confluency, 5 × 10^6^ cells were plated in 10 cm cell culture dishes (Corning, Loughborough, UK) 24 h before use. MCF7 TRCNOT1KD cells were grown under similar conditions, except tetracycline-free FCS (PAA) was used.

### 2.2. Cell Line Generation

#### Doxycycline Inducible CNOT1 Knockdown Cell Lines

To generate constructs for inducible knockdown, oligonucleotides (100 pmol of each) were phosphorylated by T4 polynucleotide kinase (50 μL reaction volume) and annealed via incubation at 95 °C, followed by slow cooling to room temperature. An aliquot (1 μL) of the duplex oligonucleotide was ligated into the pTER vector (RNAi delivery systems) [56] that had been digested with BglII and HindIII. The oligonucleotides used were as follows:

Non-targeting control (top): 5′-GATCCCtagcgactaaacacatcaaTTCAAGAGAttgatgtgtttagtcgctaTTTTTGGAAA-3′; non-targeting control (bottom): 5′-AGCTTTTCCAAAAAtagcgactaaacacatcaaTCTCTTGAAttgatgtgtttagtcgctaGG-3′.

CNOT1 (top): 5′-GATCCCcaagttagcactatggtaaTTCAAGAGAttaccatagtgctaacttgTTTTTGGAAA-3′; CNOT1 (bottom): 5′-AGCTTTTCCAAAAAcaagttagcactatggtaaTCTCTTGAAttaccatagtgctaacttgGG-3′.

To generate inducible knockdown cell lines, the pTER/control and pTER/CNOT1 plasmids were transfected into MCF7 TR cells using the calcium phosphate method followed by selection using 300 μg/mL zeocin. After expansion, MCF7 TR/control cells were selected according to their ability to knockdown transient luciferase expression after the addition of 1 μg/mL doxycycline (DOX). MCF7 TR/CNOT1 KD cells were selected via Western blotting and phenotypic analysis (reduced proliferation after the addition of 1 μg/mL doxycycline). The MCF7 TR cell line expressing the tetracycline repressor was generated via the transfection of plasmid pcDNA4/TR (ThermoFisher Scientific, Waltham, MA, USA) into MCF7 cells using the calcium phosphate method and routinely cultured in the presence of 5 mg/mL Blasticidin S. For knock-down experiments, MCF7 cells were treated with 1 µg/mL doxycycline (DOX) for the times stated.

### 2.3. Doxycycline Inducible RNase H1-DEAD U2OS Cell Lines

Polyclonal, stable inducible GFP-RNaseH1 U2OS cells were generated via lentiviral infection with pCLX-pTF-R1-DEST-R2-EBR65-RNH1-GFP lentiviral particles, followed by selection with 10 μg/mL blasticidin as described [57].

### 2.4. HEK293T/CNOT7-KO and HEK293T/CNOT8-KO Cell Lines

HEK293T CNOT7-KO and CNOT8-KO cell lines were generated using Cas9-mediated genome editing [58]. pSpCas9(BB)-2A-Puro (PX459) V2.0 was a gift from Dr Feng Zhang (Addgene plasmid #62988; http://n2t.net/addgene:62988 (accessed on 30 March 2021); RRID: Addgene_62988) (Addgene, Watertown, MA, USA). Oligonucleotides that encode small guide RNAs targeting exon 3 of the human CNOT7 gene (target sequence: TTACAGGACACCGAGTTTCCAGG) or exon 2 of the human CNOT8 gene (target sequence: GCCAGGTTATCTGTGAAGTGTGG) were inserted into PX459. The resultant PX459 vectors targeting human CNOT7 or CNOT8 were transfected into HEK293T cells (RCB2202, RIKEN BRC) using TransIT-LT1 (Mirus Bio, Madison, WI, USA). Twenty-four hours after transfection, the transfected cells were cultured in the presence of puromycin (1 μg/mL) for 2 days. The puromycin-resistant cells were then cultured from a single cell in 96-well plates. Clonal cell populations were expanded to 60 mm culture dishes and harvested for immunoblot and genome sequencing. CNOT7-KO or CNOT8-KO HEK293T cell clones validated via immunoblot and genome sequencing were used.

### 2.5. siRNA Transfection of Cell Lines

ON-TARGETplus siRNA-SMARTpools were obtained from Horizon Discovery (Dharmacon, Cambridge, UK), (Table 1); all were designed to reduce the gene expression of human CNOT1, CNOT7 and CNOT8. siRNA (0.2 nmoles) was mixed with Oligofectamine (Invitrogen, Carlsbad, CA, USA) in Opti-MEM (Gibco, Carlsbad, CA, USA) for 30 min and then added to cells in a 6 cm culture dish supplemented with 1 mL Opti-MEM. After 5–6 or 24 h of incubation at 37 °C, the Opti-MEM transfection mixture was removed and replaced with fresh DMEM medium containing 8% FCS. The cells were incubated until harvested or split. The siRNAs used in this study are detailed below, together with a control lacZ control.

### 2.6. Transient Transfection of DNA

The cells were treated with siRNA 24 h prior to DNA plasmid transfection. Appropriate CMV6-AC-RNase H1-GFP (Origene-PS100010) (2.5 μg per 6 cm dish and 1 µg per well of a 6 well plate) was mixed with Lipofectamine 2000 (Invitrogen, Carlsbad, CA, USA) in Opti-MEM, incubated for 30 min and then added to cells at 70% confluency growing in DMEM supplemented with 8%FCS. After overnight incubation at 37 °C, the Opti-MEM transfection mixture was removed and replaced with DMEM medium supplemented with 8% FCS and incubated for a further 48 h.

For the CNOT1 complementation experiments, MCF7TRCNOT1 KD cells, grown to 70% confluency, were treated with doxycycline for 48 h and then transfected with a plasmid expressing Flag-tagged CNOT1 (pcDNA3-CNOT1, GenScript, Piscataway, NJ, USA, clone ID: OHu08018) in a similar manner to CMV6-AC-RNase H1-GFP. The CNOT1 plasmid was not siRNA-resistant but the addition of large amounts of exogenous DNA led to an appreciable increase in the levels of CNOT1 protein, as shown in Figure 1H.

### 2.7. Colony Survival Assays

CNOT1 siRNA-transfected and control siRNA-transfected HeLa cells were plated at appropriate concentrations in 6 cm cell culture dishes. The cells were incubated with fresh DMEM for 2 weeks until large colonies formed. The colonies were rinsed with PBS, stained with 0.5% crystal violet in 20% ethanol, and counted.

### 2.8. Natural Comet Assay

Comet slides (X-tra^®^ Slides from Leica Biosystem, Milton Keynes, UK) were prepared by scratching with an engraving pen and then coating with 0.6% normal agarose (Sigmaaldrich, St. Louis, MI, USA) to a depth of 2–3 mm. Then, 5 × 10^5^ cells, treated with control or CNOT1 siRNA for 72 h, were suspended in 1.2% LMP agarose (Sigmaaldrich, St. Louis, MI, USA) (each sample was in triplicate) and added to the slides. After electrophoresis, the slides were rinsed with ice-cold water and incubated in 1 M Tris HCl pH 7.0 for 30 min, rinsed again in ice-cold water, heated at 42 °C overnight and then placed at 4 °C. The slides were stained with 1 mL of SYBR Green in PBS and cover slips were applied. A total of 200 comets were viewed at 20× magnification, and the mean tail moment was calculated using Open Comet plugin ImageJ bundle (original macro from Herbert M. Geller, NIH, 1997, later development by Robert Bagnell, 2011, UNC-CH).

### 2.9. Nocodazole Mitotic Shake Off

The cells were synchronized in mitosis using nocodazole (200 ng/mL) for 18 h. Nocodazole shake-off cells were centrifuged at 470× *g* for 5 min. The medium was removed, and the pellet washed once with PBS. Mitotic cells were released from the G2/M block in fresh media, sub-cultured into 6 cm dishes and were harvested 0, 1, 2, 4, 8, and 24 h later. The pellet obtained from the trypsinised cells was retained for further investigation. Cell pellets, obtained from different time points, were washed once with PBS, and lysed in urea-Tris-buffer (UTB, see below) lysis buffer (whole cell lysate) or subjected to sub-cellular fractionation.

### 2.10. FACS Analysis: Cell Cycle Analysis via Quantitation of DNA Content (Using Propidium Iodide Staining)

The cell culture medium was removed from 72 h CNOT1 siRNA-transfected and control siRNA-transfected HeLa cells. Following harvesting and ethanol fixation, the cells were rehydrated with 10 mL ice-cold PBS on ice for 30 min. The cells were washed in PBS and incubated in ice-cold 0.25% Triton-X100 in PBS at 4 °C for 15 min. After washing with 1% Bovine serum albumin (BSA) in PBS, the cells were incubated in 1 mL propidium iodide (final concentration 10 μg/mL) in PBS containing 0.1 mg/mL RNase A at room temperature in the dark for 30 min. The cells were analysed using a BD LSRFortessa™ X-20 flow cytometer (Becton Dickinson, Franklin Lakes, NJ, USA).

### 2.11. FACS Analysis: G1/S Cell Cycle Sub-Population Analysis Using EdU and BrdU Dual Pulse Labeling

A total of 72 h post siRNA transfection, the HeLa cells were pulsed with EdU (20 mM) for 2 h and then treated with CDK4/6 inhibitor palbociclib (1 μM) for 24 h leading to the arrest of 90% of the cells in G0/G1. The cells were then pulsed with BrdU (10 μM) for 2 h. Following harvesting and ethanol fixation, the cells were permeabilised using 0.2% Triton-X100 in PBS containing 2 M HCL for 30 min at RT and protected from light. The Click-iT™ EdU Alexa Fluor™ 488 Flow Cytometry Assay Kit (ThermoFisher Scientific, C10420) and Alexa Fluor 647 azide (A10277, Invitrogen) were used for the EdU staining step. BrdU was detected with mouse anti-BrdU antibody (555627BD Pharmingen BD Biosciences, Becton Dickinson, Franklin Lakes, NJ, USA). The secondary antibody was Alexa Fluor 488 goat anti-mouse (A11001, Invitrogen).

### 2.12. Metaphase Spread Analysis

CNOT1 siRNA- and control siRNA-transfected Hela cells were incubated with 0.02 µg/mL KaryoMAX^®^ Colcemid™ Solution (Gibco^®^) for 3.5 h. The cells were harvested via trypsinization and washed once in PBS, pelleted via centrifugation, and then subjected to pre-warmed hypotonic buffer (0.1 M potassium chloride) for 30 min at 37 °C to lyse the cell membranes. The cells were subsequently fixed in a fresh 3:1 ethanol–acetic acid solution. Metaphase chromosomes were immobilized on acetic-acid-humidified slides by dropping the fixed metaphase from a height of 1 m. The slides were incubated for 15 min in Giemsa (Sigmaaldrich, St. Louis, MI, USA; 5% vol/vol in water) and washed in water for 5 min. Cytogenetic analyses were performed with an Axiovert 100 (ZEISS, Birmingham, UK) inverted microscope.

### 2.13. Co-Immunoprecipitation

HeLa cells were washed in ice-cold PBS and lysed in NETN buffer (0.15 M NaCl, 40 mM Tris, 5 mM EDTA, pH 7.4) via homogenization using a Wheaton-Dounce hand homogenizer. Lysates were clarified via centrifugation at 44,000 rpm for 30 min and then incubated with appropriate antibodies overnight. Antibody–antigen complexes were isolated using Protein G-agarose beads (Generon, Slough, UK) for 2 h. After washing with NETN buffer, the beads were heated in SDS sample buffer, and the released proteins were fractionated using SDS.PAGE followed by Western blotting to identify co-immunoprecipitating proteins. An antibody against collagen V was used as a negative control.

### 2.14. DNA Fibre Analysis

After appropriate treatment with control or CNOT1 siRNA for 72 h, the cells were pulse-labelled with 25 μM CldU and 250 μM IdU for 20 min and harvested. DNA fibre spreads were prepared by spotting 10^3^ cells onto microscope slides followed by lysis with 7 μL 0.5% SDS, 200 mM Tris-HCl pH 7.4 containing 50 mM EDTA. The DNA spreads were fixed in methanol–acetic acid (3:1) and then treated with HCl. After washing with PBS, HCl-treated fibre spreads were incubated with rat anti-bromodeoxyuridine (detects CldU, AbD Serotec) and mouse anti-bromodeoxyuridine (detects IdU, Becton Dickinson, East Rutherford, NJ, USA) antibodies for 1 h, fixed with 4% paraformaldehyde (PFA) in PBS to increase staining intensity and incubated with anti-rat IgG AlexaFluor 555 and anti-mouse IgG AlexaFluor 488 antibodies (Molecular Probes) for 1.5 h. The images were acquired using a Nikon E600 microscope (Nikon, Tokyo, Japan) with a Nikon Plan Apo × 60 (1.3 numerical aperture) oil lens, a Hamamatsu digital camera (C4742-95) and the Volocity acquisition software (version v4.1) (Perkin Elmer, Milan, Italy). The images were analysed using ImageJ; in each independent experiment, at least 300 fibres were measured per condition.

### 2.15. Immunofluorescence Microscopy

The cells were washed and fixed as follows. For 53BP1 staining, the cells were treated with CSK buffer (10 mM PIPES, 300 mM sucrose, 100 mM NaCl and 3 mM MgCl_2_) for 1 min, then with 0.5% Triton X-100 in CSK buffer for 1 min, and finally with 4% PFA in PBS for 10 min at room temperature; for Cyclin A, the γH2AX and CldU staining cells were treated with 4% PFA for 10 min and 0.3% Triton Χ-100 in PBS for 5 min at room temperature; the S9.6 staining cells were treated with methanol for 10 min on ice and 0.5% Triton X-100 in PBS for 5 min at room temperature. The samples were blocked with 3% BSA/10% foetal bovine serum in PBS. Primary antibodies were mouse anti-phospho-Histone H2AX (Ser139) (JBW301, Millipore 05-636, 1:1000), rabbit anti-53BP1 (Bethyl A300-272A, 1:30,000), mouse anti-Cyclin A (6E6, Thermo Scientific MS1061, 1:50), rat anti-CldU (BU1/75, AbD Serotec OBT0030G, 1:250) and mouse anti-RNA/DNA hybrid (S9.6, gift from Professor Richard Gibbons, hybridoma supernatant 1:100). The secondary antibodies were anti-mouse IgG AlexaFluor 488 and anti-rabbit IgG AlexaFluor 555 MolecularProbes). DNA was counterstained with 4,6-diamidino-2-phenylindole (DAPI) and images acquired as above. For the quantification of nuclear S9.6 intensity, ImageJ was used to generate nuclear masks based on DAPI staining and mean S9.6 fluorescence intensities per pixel were quantified per nucleus.

### 2.16. EU Incorporation Assay

EU incorporation assays were performed using the Click-iT RNA Alexa Fluor 594 Imaging Kit (Invitrogen) according to the manufacturer’s instructions. The cells were incubated with 1 mM EU in DMEM for 1 h, fixed with 4% PFA in PBS for 15 min at room temperature, permeabilized with 0.5% Triton X-100 in PBS for 15 min and the Click-iT reaction performed. DNA was counterstained with DAPI, and the images were acquired as above. ImageJ was used to generate nuclear masks based on DAPI staining and mean AlexaFluor 594 fluorescence intensities per pixel were quantified per nucleus.

### 2.17. Western Blotting

The cell extracts were prepared in UTB buffer (50 mM Tris-HCl pH 7.5, 150 mM β-mercaptoethanol and 8 M urea) and sonicated to shear DNA and to release DNA-bound proteins. After protein concentration determination, the cell lysates were fractionated via SDS.PAGE and the proteins were electrophoretically transferred to nitrocellulose membranes. Membranes were incubated in diluted antibody overnight at 4 °C. After washing in Tris-buffered saline containing 0.1% Tween 80, the membranes were incubated in HRP-linked anti-mouse or anti-rabbit antibody for 2 h as appropriate. The antigens were visualised using ECL reagent (GE Healthcare, Chicago, IL, USA) and X-ray film (Kodak, Rochester, MN, USA). The primary antibodies used in this study are listed in Table 2.

Location of suppliers of antibodies is as follows: Proteintech, Rosemont, PA, USA; Santa Cruz, Dallas, TX, USA; Novus, Abingdon, UK; Millipore, Burlington, MA, USA; Cell Signalling, Danvers, MA, USA; Becton Dickinson, Franklin Lakes, NJ, USA; Abcam, Cambridge, UK; Bethyl Laboratories, Montgomery, AL, USA; R and D Systems, Minneapolis, MN, USA.

### 2.18. Slot-Blot Experiments

Genomic DNA (1.2 μg) was isolated and treated with 2 units of RNase H per μg of DNA (NEB, M0297S) for 2 h at 37 °C before being loaded onto the slot blot apparatus. Half of the DNA sample was blotted onto a nitrocellulose membrane probed with S9.6 antibody (1:1000) and the other half probed with an anti-ssDNA antibody (MAB3031, Millipore, Milan, Italy, 1:5000) as above as a loading control.

### 2.19. Quantitative Real-Time PCR

Total RNA was harvested using TRIZOL reagent (Invitrogen) followed by DNase I treatment (Roche, Rotkreuz, Switzerland); 1.5 μg of total RNA was reverse transcribed using SuperScript Reverse Transcriptase III (Invitrogen) with random hexamers (Invitrogen) following the manufacturer’s instructions. The qPCR TBP forward and reverse primers for amplification are listed below. For quantitative real-time PCR, 2 μL of cDNA was analysed using a Rotor-Gene RG-3000 real-time PCR machine (Corbett Research, Sydney, Australia) with QuantiTect SYBR green (Qiagen, Manchester, UK). The cycling parameters were 95 °C for 15 min, followed by 45 cycles of 94 °C for 20 s, 58 °C–62 °C for 20 s and 72 °C for 20 s. TBP (F) TTC GGA GAG TTC TGG GAT TG TBP. (R) CTC ATG ATT ACC GCA GCA AA.

### 2.20. Total RNA-Sequencing

HeLa cells were treated with control or CNOT1 siRNA for 72 h. After harvesting, total nucleic acid was extracted using a FlexiGene Kit (Qiagen, Manchester, UK). The quality and quantity of extracted RNA were measured using a Qubit 2.0 Fluorometer (Life Technologies, Carlsbad, CA, USA) and an Agilent 2200 TapeStation system, respectively. RNA concentrations and purity were measured in pg/μL and RIN (RNA Integrity Number), respectively, and the results were recorded. RIN 7 was set as the cut-off value for sequencing (the highest score for RIN is 10). The 28S:18S rRNA ratio was also calculated as a marker for RNA quality. A 28S:18S rRNA ratio of 2:1 was considered a representative of good-quality RNA. A dual-indexed, strand-specific RNA-Seq library was prepared from submitted total RNA, using RiboZero (Illumina, San Diego, CA, USA) rRNA depletion and the NEBNext Ultra Directional RNA library preparation kit (NEB). RNA-Seq was performed on 2 lanes of HiSeq4000 (paired-end, 2 × 150 bp) platform to generate ~80 M reads per sample. The overall alignment rate for most of the samples was above 90%. FDR < 0.05. The raw data files were generated in binary base call (BCL) format and converted to FASTQ format using bcl2fastq Conversion Software v2.20 (Illumina). FASTQ format readings were transferred to BaseSpace^®^ Sequence Hub (Illumina). The assay was run in 4 lanes in paired-end and 76-cycle mode. The FASTQ files were aligned using HISAT2 [59], and the resulting mRNA transcripts were assembled using StringTie [60] as described in [61]. All RNA-Seq data were uploaded to the GEO database (https://www.ncbi.nlm.nih.gov/geo/query/acc.cgi?acc=GSE141496 accessed on 4 December 2019) with the accession number GSE141496.Taking the control siRNA sample as reference, the log2-transformed fold changes (log2FC) of CNOT1 siRNA samples were calculated. Genes with positive log2FC were considered “upregulated”, while those with negative log2FC were considered “downregulated”. To determine whether the change in the expression of a particular gene was significant, we used empirical *p*-values. After correcting the *p*-values for false positive rates in multiple testing problems, using the false discovery rate as proposed in the work of [62], we analysed the enrichment of the up-/downregulated gene sets for each sample using Enrichr [63]. Finally, significant biological processes reported by Enrichr were extracted for further consideration.

## 3. Results

### 3.1. CNOT1-Depleted Cells Show Increased Global Transcriptional Activity

Previous chromatin immunoprecipitation (ChIP) analysis in control (CNOT1fl/fl) and CNOT1-Liver KO mice showed increased RNAPII occupancy on genomic regions of the immune system’s process- and apoptosis-related genes in the absence of CNOT1, leading to the increased expression of mRNAs for transcription factors, cell cycle regulators, and DNA damage response-related proteins, as well as apoptosis-related and inflammation-related genes [64]. To determine whether there was an increase in transcription in human cells, HeLa cells were treated with CNOT1 siRNA and control siRNA, and the efficiency of the depletion was confirmed via Western blotting up to 120 h (Figure 1A). Based on this result, the 72 h post CNOT1 siRNA treatment was used for most of the subsequent experiments. The expression level of the other CNOT subunits was analysed in control and CNOT1-depleted HeLa cells 72 h post siRNA transfection. The depletion of CNOT1 leads to a reduction in the level of most other CNOT subunits except CNOT4 (Figure 1B) and reduction in deadenylase activity (Appendix A). An explanation for this could be that mammalian CNOT4, unlike Not4, which is constitutively present in the yeast complex, is not stably associated with the other subunits in the mammalian complex but can associate with the complex weakly [65]. However, the association of Not4, or its E3 activity with the complex, has been shown to be tightly controlled [66,67]. In addition, the association of CNOT4 with the full human CNOT complex is required for optimal deadenylase activity [67].

The effect of CNOT1 depletion (Figure 1C) on transcription activity was assessed by quantifying the level of RNA-specific labelled nucleoside 5-ethynyl uridine (EU) incorporated into nascent RNA (Figure 1D). RNA synthesis (EU incorporation) was elevated 48 h post CNOT1 depletion and was slightly decreased on day 3 but was increased further on day 4; this incorporation was significantly greater than the level seen in control cells (Figure 1E). An increased level of transcription activity due to the activation of nucleolar RNAPII in CNOT1-depleted cells was also indicated by the Western blotting results, which showed increased expression of TBP and increased phosphorylation of RNAPII C-terminal domain at serine 5, a mark of transcriptional activation, compared to the control cells (Figure 1F).

In a confirmatory experiment, we investigated the global transcription level in MCF-7 cells depleted of CNOT1. MCF7 TR cells were routinely treated with 2 µg/mL DOX for 3 days to deplete CNOT1; these conditions were used for most subsequent experiments. Similar to CNOT1-depleted HeLa cells following knocking down of CNOT1, the level of most other CNOT subunits except CNOT4 and CNOT6 were reduced (Figure 1G). The rescue assay was performed in MCF7 cells using flag-tagged CNOT1 pcDNA3 transfection for 48 h (Figure 1H–J). The global transcription level increased in MCF7 TR/CNOT1 KD cells 3 days post 2 μg/mL DOX treatment compared to MCF7 TR/Control cells (Figure 1K,L). The re-expression of CNOT1 reduced the elevated levels of EU incorporation/intensity in MCF7 TR/CNOT1 KD; however, it was not statistically significant, probably due to low transfection efficiency, which was approximately 20% (Figure 1K,L). Up to 5 days post 2 μg/mL DOX, inducible MCF7 TR/ CNOT1 KD cells showed a gradual upregulation of TBP expression and increased phosphorylation of the RNAPII C-terminal domain on serine 5 (Figure 1M).

RNA-Seq analysis (discussed in detail below) did not show TBP transcript upregulation and this upregulation was only observed at the protein level. However, TBP mRNA stability was increased up to 6 h post DRB treatment in MCF7 TR/ CNOT1 KD cells compared to control cells, which could explain its upregulation at both mRNA and protein levels (Appendix A).

### 3.2. Depletion of CNOT1 Leads to Increased R-Loop Formation

Transcription may affect genome integrity if co-transcriptional products such as RNA–DNA hybrids (R-loops) are not efficiently removed behind RNA Polymerase II [68]. Therefore, we investigated whether an elevated transcription activity in CNOT1-depleted/knocked down cells leads to increased R-loop formation. To examine this, slot blot analysis of gDNA isolated from siRNA control and CNOT1-depleted Hela cells was performed. gDNA samples were transferred to nitrocellulose membranes using the slot blot apparatus. After cross-linking, the membranes were incubated with S9.6 antibody, which detects RNA–DNA hybrids, or an antibody against single-strand DNA (ssDNA). R-loops were significantly enriched in CNOT1-depleted cells compared to cells transfected with control siRNA. The S9.6 signal was diminished in the samples after pre-treatment of gDNA with RNase H1 enzyme for 2 h, demonstrating the signal derived from RNA (Figure 2A). In a further experiment, the intensity of nuclear S9.6 signal was quantified in HeLa (Figure 2C,D) and MCF-7 TRCNOT1KD (Figure 2E,F) cells via confocal microscopy. The efficiency of siRNAs in the depletion of CNOT1 expression was confirmed via Western blotting (Figure 2B). Additionally, the rescue assay was performed in MCF7 cells using flag-tagged CNOT1 pcDNA3 transfection for 48 h (Figure 2G,H). Fixed and permeabilized cells were co-labelled with S9.6 antibody and an antibody against the nucleolar protein nucleolin to enable the demarcation of the nucleoli [69]. The level of S9.6 signal was significantly higher in CNOT1-depleted HeLa cells and MCF7 TR/CNOT1 KD cells compared to their control conditions. The level of S9.6 signal was decreased after transfection of HeLa cells with RNase H1 for 24 h. The reduction was greater in control HeLa cells compared to CNOT1-depleted cells (Figure 2D). The ability of endogenous RNase H1 enzyme to remove R-loops from gDNA was somewhat more efficient than the transiently transfected recombinant RNase H1 (Figure 2C,D). This could be due to the presence of RNase H-resistant hybrids or incomplete nuclease activity in the nucleolus, which is rich in DNA-RNA hybrids. It is also likely that low transfection efficiency of the construct contributed to this.

The level of S9.6 signal was decreased, almost to the same level as control cells, in MCF7 TR/CNOT1 KD cells 48 h post transfection with CNOT1 pcDNA3 (Figure 2F). In order to investigate the role of deadenylase subunits in the formation of RNA–DNA hybrids, we measured the level of S9.6 signal in CNOT7-KO and CNOT8-KO HEK293T cells. In addition to these cells, we also used control and CNOT1 siRNA-treated HEK293T cells as a control. As expected, the depletion of CNOT1 significantly increased the level of S9.6 signal in HEK293T cells compared to the control cells (Appendix A). Both CNOT7-KO and CNOT8-KO HEK293T cells showed a significant upregulation of S9.6 signal but at a lower level than CNOT1-depleted HEK293T cells (Appendix A). The difference between the results obtained for CNOT1 depletion and knock down of each deadenylase subunit is probably because CNOT7 and CNOT8 are partially redundant [70]. Similarly, the CNOT6 and CNOT6L deadenylase subunits were unaffected in the HEK293T experiment.

Since the S9.6 antibody binds to double-stranded RNA (dsRNA) as well as RNA–DNA hybrids in vitro and in vivo, giving rise to nonspecific signals, in a separate experiment, we measured the R-Loop formation in U2OS cells stably expressing GFP-tagged catalytically active and inactive RNase H1 protein. GFP-dead (dRNH1) U2OS cells were transfected with control siRNA, CNOT1 siRNA, CNOT4 siRNA or double CNOT7 and CNOT8 siRNAs. The efficiency of the depletion was confirmed via Western blot analysis (Figure 2K,L). CNOT1 depletion significantly elevated the GFP signal in dRNH1compared to that in the control (Figure 2I). This increase was even greater in GFP-dRNH1 U2OS cells compared to GFP-wtRNH1 U2OS cells, confirming that RNaseH1 exhibits specific activity toward RNA–DNA hybrid removal. Double siRNA depletion for CNOT7 and CNOT8 slightly increased the GFP signal in dRNH1 U2OS cells compared to control. The depletion of CNOT4 showed no difference compared with the control sample in GFP-dRNH1 U2OS cells (Figure 2I,J).

### 3.3. Depletion of CNOT1 Induces Replication Stress through Ongoing Transcription

As previously reported, increased global transcription activity can lead to elevated replication stress and genome instability [40]. To test the hypothesis that an increase in transcription observed in CNOT1-depleted cells may underlie replication stress, RNA synthesis was inhibited using 100 µM 5,6-dichloro-1-beta-D-ribofuranosylbenzimidazole (DRB) for 100 min. The effect of DRB on RNA polymerase II is due to CDK9 inhibition. The CDK9 is a core component of P-TEFb by binding with cyclinT1 to form an active P-TEFb kinase, which is a complex required for processive transcription elongation by RNA polymerase II [71,72]. Treatment with DRB significantly reduced the nuclear EU intensity in both control and CNOT1-depleted cells, which is consistent with the decreased levels of RNA synthesis (Figure 3A,B). In a further experiment, the effect of CNOT1 depletion on DNA replication was examined directly. Using DNA fibre analysis, DNA replication fork speeds were determined in the absence of CNOT1. The depletion of CNOT1 was seen to reduce fork speeds appreciably in Hela cells (Figure 3E–G). However, in the presence of DRB replication, the fork speeds increased specifically in CNOT1-depleted cells (Figure 3F). Similarly, treatment with PHA-767491 hydrochloride significantly rescued replication fork progression, presumably via a reduction in replication initiation or transcription elongation through the inhibition of CDC7 or CDK9, respectively, in CNOT1-depleted cells (Figure 3G). These data suggest that replication stress in CNOT1-depleted cells is associated with increased active RNA synthesis. The efficiency of siRNAs in the depletion of CNOT1 expression was confirmed via Western blotting (Figure 2C,D).

Similarly, the knock down of CNOT1 in MCF7 TR CNOT1KO cells significantly reduced DNA fork speeds, while the re-expression of CNOT1 rescued this phenotype as the peak was shifted from left to right, confirming the specific role of CNOT1 in the regulation of DNA replication (Figure 3J,K). The rescue assay was performed in MCF7 cells using flag-tagged CNOT1 pcDNA3 transfection for 48 h (Figure 3H,I). Moreover, a series of co-immunoprecipitation (Co-IP) assays was conducted with a panel of pre-replication complex (pre-RC) antibodies (MCM2, MCM7 and ORC3). The Co-IP assays were carried out using HeLa cell lysates and revealed that MCM2, MCM7 and ORC3 associated with CNOT1 and TAB182 (a closely associated or integral component of the mammalian Ccr4–Not complex), suggesting that the Ccr4–Not complex may interact with the pre-RC at DNA replication origins (Appendix A).

### 3.4. Depletion of CNOT1 Induces Replication Stress through R-Loop Formation

To investigate the contribution of R-loops to replication fork slowing in CNOT1-depleted cells, recombinant RNase H1-GFP was transfected into CNOT1-depleted and siRNA-transfected control HeLa cells. RNase H1-GFP did not have a significant effect on replication initiation (origin firing) (Figure 4B); however, it rescued replication fork progression in CNOT1-depleted cells, although to a limited extent, but not in control cells. This observation supports the suggestion that replication stress in CNOT1-depleted cells is promoted via increased R-loop formation following increased RNA synthesis (Figure 4C). The efficiency of siRNAs in the depletion of CNOT1 expression was confirmed via Western blotting (Figure 4A).

### 3.5. Increased Level of Double-Strand Breaks in Cells Lacking CNOT1

In view of the observation that the depletion of CNOT1 leads to increased R-loop formation and DNA replication stress, we next assessed to what extent this results in DNA damage. CNOT1 protein is present in the nucleus but does not co-localize with γ-H2AX foci at the site of DNA damage following IR (3 Gy) treatment (Appendix A). However, CNOT1 depletion yielded a statistically significant increase in the proportion of cells with >10 γ-H2AX foci in MCF-7 TRCNOT1KD cells (Figure 5A,B).

The level of DNA damage in control and CNOT1-depleted HeLa cells and MCF7 TR/CNOT1 KD cells was measured using different approaches. In the first set of experiments, the number of cells with single and multiple micronuclei per cell were separately counted in both HeLa and MCF-7 cells after the depletion of CNOT1. The number of cells with multiple micronuclei was found to be significantly elevated in CNOT1-depleted HeLa cells (5 days post siRNA transfection) (Figure 5H,I) and MCF7 TR/CNOT1 KD cells (3 days post +/− 2 μg/mL DOX treatment) (Figure 5C,D). In addition, a significant increase in 53BP1 nuclear body formation was observed in G1 positive CNOT1-depleted HeLa cells (Figure 5J,K) and MCF7 TR/CNOT1 KD cells (Figure 5E,F).

In further experiments, neutral comet assays showed longer Olive tail moments in CNOT1-depleted HeLa cells, confirming increased damage formation 120 h post siRNA transfection (Figure 5L). CNOT1siRNA-transfected and control siRNA-transfected Hela cells were synchronized in metaphase with Colcemid™ Solution, harvested and then subjected to incubation in pre-warmed hypotonic buffer. Metaphase chromosomes immobilized on acetic-acid-humidified slides were stained with Giemsa. Chromosomal aberrations were assessed in control and CNOT1-depleted HeLa cells in the presence of colcemid using metaphase spreads (Figure 5M). The cells lacking CNOT1 exhibited higher numbers of gaps/breaks and radials compared to the other conditions (Figure 5N). One is left wondering if the observed phenotypes are a direct consequence of CNOT1 loss or due to subsequent disruption of the complex and its associated enzymatic activities. Therefore, we have considered the effects of knocking down two of the deadenylase adaptor subunits (CNOTs 7 and 8). This has been compared to the effect of knocking down CNOT4 and CNOT1 in HEK293T cells (Appendix A). The percentage of cells with >10 γ-H2AX foci and micronuclei formation per cell was significantly increased in CNOT1 knock down, CNOT7 KO and CNOT8 KO HEK293T cells compared to the control cells (Appendix A). However, CNOT4 knock down cells showed no difference compared with the control cells (Appendix A). These results strongly suggest that the loss of deadenylase activities of the complex contributes to the observed phenotypes. (As shown in Appendix A, the depletion of CNOT1 results in a reduction in deadenylase activity of the complex.).

### 3.6. CNOT1 Depletion Leads to Activation of the ATM/Chk2 and Downregulation of the ATR/Chk1 Repair Pathways

The ATR-Chk1 axis stabilizes stalled replication forks locally, suppresses origin firing globally and prevents fork collapse into DNA double-strand breaks (DSBs) [73]. Here, we show that DNA fork collapse into DSBs in CNOT1-depleted HeLa cells is associated with the downregulation of the ATR/Chk1 signalling pathway (Figure 6A). The level of DNA single-strand breaks (SSBs) was assessed by measuring the percentage of HeLa cells with >10 RPA foci (Figure 6B). This level was slightly lower in CNOT1-depleted cells compared to the control cells (Figure 6C). However, it has been shown that ATR-deficient cells undergo nucleus-wide breakage after unscheduled origin firing generates an excess of SSBs that exhausts the nuclear pool of RPA (called “RPA exhaustion”), which leads to a rapid conversion of the SSBs to DSBs, which, in turn, causes replication catastrophe and cell death [74]. Therefore, we were not able to confirm that the reduction in RPA foci formation in CNOT1-depleted cells is a true reflection of reduced SSB level or is due to RPA exhaustion. A CRISPR screen revealed that CNOT1 deficiency was synthetically lethal with ATR inhibition (ATRi AZD6738) both in vitro and in vivo [75], thus confirming they work in parallel, mutually compensatory pathways that, together, perform an essential function.

Using a co-IP assay, we showed that ATR associates with CNOT1 and Tab182 in human cells, possibly indicating an interaction with the intact Ccr4–Not complex (Appendix A). We have also shown that phosphorylation of the effector kinase Chk1 at serine 345 and at RPA serines 4/8 were reduced following CNOT1 depletion in HeLa cells (Figure 6A). These are considered usual ATR substrates. We have also shown that the expression of ATR and its binding partner ATRIP were both down regulated in CNOT1-depleted cells, 72 h post siRNA transfection (Appendix A). However, no difference was observed in the level of the ATR activator protein TopBP1 between the control and the CNOT1-depleted cells. It also appears that ATR loading onto chromatin was reduced in the absence of CNOT1 (Appendix A).

The ATM/Chk2 signalling pathway is activated in response to an increase in DSBs formation. We have shown activation of ATM/Chk2 repair pathway in both CNOT1-depleted HeLa cells and MCF7 TR/CNOT1 KD up to 4 days post siRNA transfection and 3 days post DOX treatment, respectively (Figure 6D,E). The longer time points were used to allow the CNOT1-mediated DSBs to become detectable and trigger the threshold damage response. Increased phosphorylation of the ATM (S1981) and its substrates H2AX, Chk2 and KAP1 (S139, T68 and S824, respectively) was also observed in CNOT1-depleted HeLa cells and MCF7 TR/CNOT1 KD using Western blotting analysis (Figure 6E).

### 3.7. DNA Damage-Induced Cell Cycle Arrest and Senescence in CNOT1-Depleted Cells

To better understand the mechanisms through which the depletion of CNOT1 results in increased transcription, an RNA-Seq analysis was performed. HeLa cells were treated with control and CNOT1 siRNA for 72 h. The cells were harvested and processed for RNA-Seq as described in the Section 2. The results have been deposited in the GEO database https://www.ncbi.nlm.nih.gov/geo/query/acc.cgi?acc=GSE141496 (accessed on 4 December 2019) with the accession number GSE141496. An MA plot for differential gene expression between control siRNA vs. CNOT1 siRNA-treated HeLa cells was also generated (Appendix A).

The upregulation of genes involved in the repair of damaged DNA, activation of the immune system, inflammatory response, and differentiation/transformation of epidermal cells was evidenced by data collected from RNA-Seq following the downregulation of CNOT1 (Figure 7A,B). In addition, the genes involved in the activation of transcription factor regulator ERK1/2 were also significantly upregulated following the downregulation of CNOT1 (Figure 7A,B). For instance, the *HTR2B* gene encodes one of the several receptors for 5-hydroxytryptamine 5-HT2B (serotonin) that belongs to the G-protein-coupled receptor 1 family. The phosphorylation of ERK1/2, is a known target of 5-HT2B receptor stimulation in astrocytes [76,77]). Previous studies have shown, using promoter recruitment assays, that CNOT1 represses the ligand-dependent transcriptional activation function of oestrogen receptor (ER) α [78]. In contrast, ligand-bound ERα can be recruited to target genes via other transcriptional activators, such as activator protein 1 (AP-1), which is regulated by mitogen-activated protein kinase (MAPK/ERK1/2) signal transduction pathways [79]. Therefore, we hypothesized that in the absence of the inhibitory function of CNOT1 on transcription initiation, the activation pathway AP-1-ERK1/2 would be dominant.

To determine whether ERKs are activated in HeLa cells in the absence of CNOT1, we examined the level of phospho-ERK1/2 in control and CNOT1-depleted HeLa cells up to 72 h post siRNA transfection. In both the control and CNOT1-depleted cells, the level of phosphorylation was significantly upregulated up to 24 h, probably because the cells were growing exponentially. As soon as the cell growth reached a plateau at 48 h, the level of phospho-ERK1/2 dramatically decreased and stayed at a very low level beyond 48 h in the control cells. However, at day 3 in the CNOT1-depleted cells, a time point at which we normally observed optimal CNOT1 depletion, the level of phospho-ERK1/2 was increased, which is consistent with the RNA-Seq results (Figure 7C). To investigate whether the activation of ERK1/2 was independent of serum stimulation, in a separate experiment, 48 h post siRNA transfection, we also stimulated the phosphorylation of ERK1/2 by culturing HeLa cells in DMEM without serum for 24 h prior to stimulation with 10% FCS. The cells were harvested at 0, 1, 4 and 24 h post stimulation. A total of 4 h after treating the cells with serum, the level of phosphorylation of ERK1/2 and Akt increased before decreasing at 24 h in the control cells; however, the level of phosphorylation of both proteins was consistently high even in untreated CNOT1-depleted HeLa cells, and 24 h later when the effect of serum stimulation diminished, suggesting that ERK1/2 activation was CNOT1-dependent (Figure 7D). The constitutive activation of the ERK1/2 mitogen-activated protein kinase pathway promotes the transcription and expression of genes involved in G1 progression and cell proliferation [80]). However, hyperactivation of the pathway leads to G1 cell cycle arrest due to long-term p21 induction and CDK2 inhibition [81,82,83,84,85]. ERK1/2-induced cell cycle arrest is generally associated with the upregulation of p53, p21 and p16 [80]. In contrast to MYC, the hyperactivation of oncogenes such as RAS and RAF, ERK1/2 (MAPK), and E2F1 induces a G1 cell-cycle arrest and cellular senescence [86]. The increased expression of p53, p21 and p16 proteins in CNOT1-depeleted cells was confirmed via Western blotting (Figure 7E).

The cell cycle profile was assessed, using FACS analysis, of control and CNOT1-depleted HeLa cells 72 h post siRNA transfection. Higher numbers of cells in G1 were observed in CNOT1-depleted cells compared to controls (Figure 7F,G). In addition, we investigated the progression of the cell cycle using EdU (APC-A) and BrdU (Alexa Fluor™ 488) dual pulse labelling. The cells were synchronized in G1 using CDK4/6 inhibitor palbociclib (1 μM) for 24 h leading to 90% G0/G1 cells between BrdU and EdU labelling. The cell cycle progression of cells was analysed via FACS analysis at 0, 2, 4, 8 and 10 h post second labelling. As expected, the speed of cell cycle progression was reduced in CNOT1-depleted cells, with cells gradually accumulating in the G1/S border ($BrdU^{+}EdU^{-}$) at the 10 h time point, while the control cells had already entered the S phase by this time (Appendix A). The impact of the depletion of CNOT1 on cell cycle progression was further assessed by synchronizing the cells in mitosis using nocodazole (200 ng/mL), which blocks cells in pro-metaphase by inhibiting the mitotic spindle formation, for 18 h, 72 h post CNOT1 depletion. The dominant cell population in CNOT1-depleted cells consisted of adherent interphase cells, suggesting that the progression into and through mitosis is inhibited in CNOT1-depleted cells. Control and CNOT1-depleted mitotic cells were collected via shake-off and released from the G2/M block by washing with fresh media. The cells were harvested at the indicated times and analysed via Western blotting using the antibodies against cyclins D1 (G1), E (G1/S), A (S), B1 (G2/M) and phospho-H3 (M) (Appendix A). Strikingly, high-level expression of cyclin E, which is responsible for cell cycle transition from G1 to S phase, was seen in CNOT1-depleted cells up to 24 h after nocodazole release. Cyclin D1 was weakly expressed in CNOT1-depleted cells compared to the controls; these cells were considered to be in G1 phase. The expression of cyclins A and B1 was reduced in G2/M CNOT1-depleted cells and in the following cycles. Moreover, in CNOT1-depleted cells, the phosphorylation of the metaphase marker histone H3 (pH3) phosphorylated on serine 10 was lost 2 h after removal of nocodazole. No further evidence of H3 phosphorylation was observed even at later time points. It is also worth noting that the CNOT1-depleted cells appear to progress out of mitosis more rapidly than the controls, as evidenced by the absence of pH3 after 1 h in normal media; pH3 can be seen in control cells up to 4 h.

Based on these results, we propose the repertoire of Ccr4–Not complex mRNA homeostasis to include a function in control of cell cycle progression in mammalian cells. Indeed, it has already been shown that the Ccr4–Not complex is involved in the stability of cell cycle-associated mRNAs such as CLB2 (cyclin B human orthologue) [87], cyclins CLN2 (cyclin E human orthologue) and CLB5 (cyclin A2 human orthologue), histone H2A.1 (H1A human orthologue) [88], DBF2 and SW15 [89] in *Saccharomyces cerevisiae*.

We also considered that the arrest in G1/S could lead to the senescence of the CNOT1-depleted HeLa cells. Induction of senescence was confirmed using SA-β-gal staining at 3-, 5- and 7-days post CNOT1 siRNA transfection. Based on the results obtained in the knock-down efficiency assay and knowing that the induction of senescence without drug administration is not an immediate response, we processed samples from 72 h post siRNA treatment for both control and CNOT1-depleted HeLa cells. At later times, the CNOT1-depleted HeLa cells appeared to senesce (increased senescence-associated β-galactosidase activity) whereas the control cells did not (Figure 7J,K). Similar results reported by Mittal et al., 2011 have suggested that the knockdown of CNOT6 and CNOT6L in MCF7 cells causes the upregulation of insulin-like growth factor-binding proteins (IGFBPs), which may mediate the inhibition of cellular proliferation and induce senescence via a p53-dependent pathway [90].

In addition, we have shown that the depletion of CNOT1 significantly reduced cell growth (measured as colony formation) in CNOT1-depleted cells measured 14 days after the plating of cells in the absence of DNA-damaging agents (Figure 7H,I).

### 3.8. RNA-Seq Analysis of CNOT1-Depleted HeLa Cells

To better understand the mechanisms through which the depletion of CNOT1 results in increased transcription, an RNA-Seq analysis was performed as outlined in the introduction to the previous section. The MA plot demonstrated that there were more upregulated genes (808) genes than downregulated genes (487) following the depletion of CNOT1 in HeLa cells (Appendix A). Data collected from RNA-Seq also showed, following the depletion of CNOT1, a significant downregulation of genes involved in cytoplasmic mRNA turnover (CNOT1), peptidyl-arginine methylation (PRMT6), argininosuccinate lyase (H7C0S8), protein deSUMOylation (SENP3-E1F4A1), amino sugar metabolic processing (SLC35A3), the regulation of stem cell population maintenance and the negative regulation of retinoic acid receptor and intracellular oestrogen (ER) receptor signalling pathways (Figure 7B). Previous ChIP analysis indicated that Ccr4–Not, Mbf1p, SAGA, SWI/SNF, SRB/MED, RSC, and the Paf1 complex are all recruited by Gcn4p to one of its target stress-regulated genes Arg1 in vivo [31,33]. This may have relevance to the downregulation of the arginine methyltransferase 6 (PRMT6) gene observed here. Noteworthy, the Ccr4–Not complex was previously shown to physically interact with PRMT1, regulating its methyltransferase activity [91,92].

The upregulated genes were mostly involved in the activation of inflammatory and immune responses, disruption of cell membranes (tissue damage) and cell movement. The last two may result from a direct involvement of autophagy. Evidence in support of the role of the Ccr4–Not complex in the regulation of autophagy in yeast has shown that Not1/CNOT1 interacts with Dhh1, a component of the P-bodies, or its the mammalian homolog DDX6, and stimulates its ATPase activity. Dhh1 is a post-transcriptional repressor of autophagy, which acts by, at least partly, binding and inhibiting the expression of a set of ATG genes, including ATG3, ATG7, ATG8, ATG19, ATG20, ATG22, and SNX4/ATG24 [79]. The loss of CNOT1 or CNOT3 results in autophagy and cell death in mouse cardiomyocytes [93]. CNOT3 binds to the poly (A) tail of ATG7 mRNA and regulates its stability and ultimately level of protein expression. In hearts with depleted CNOT3, the expression of ATG7 was elevated [93]. The depletion of CNOT2 in H1299 cells leads to the induction of the autophagy adaptor protein p62/SQSTM1 and LC3B-II conversion, with impaired autophagic flux [94]. Altogether, CNOT1 suppression results in an increase in immune-related and inflammation-related genes; in contrast, the genes involved in cellular stress response become less abundant.

## 4. Discussion

The Ccr4–Not complex was originally recognized as an important regulator of gene expression in yeast. However, a possible role for the complex in the control of genome stability in the context of gene transcription has not previously been considered in detail. To address this question, we decided to examine the effects of disabling the whole complex by ablating CNOT1 expression. This has the effect of destabilizing the expression of most other components and reducing the deadenylase activity of the complex.

We have now shown that the inactivation of the Ccr4–Not complex can lead to transcription-associated replication stress. Taken together, our data demonstrate that the depletion of CNOT1 increases RNA synthesis in HeLa and MCF-7 cells judged by EU incorporation. Importantly, the treatment with transcription inhibitors DRB and PHA significantly increased the replication fork speed in CNOT1-depleted cells, suggesting that the replication stress in the absence of CNOT1 is, at least partially, due to enhanced RNA synthesis. In addition, the level of general transcription factor TBP and phosphorylation of RNA Pol II (C-terminus) at serine 5 were both upregulated in CNOT1-depleted cells, 72 h post siRNA transfection or DOX treatment, which is consistent with increased transcription. The association of TBP with increased transcription, R-loop formation, and replicative stress has been reported [40]. RNA-Seq analysis did not show TBP gene upregulation, with upregulation only being observed at the protein level. Previously, unspliced TBP mRNA (pre-mRNA) level was shown to be reduced in liver-specific CNOT1-KO mice (CNOT1-LKO) compared to control cells; in contrast, its relative mRNA expression level was significantly higher than that in control cells [64]. This shows either a quicker pre-mRNA processing (splicing) or longer mRNA lifespan (increased mRNA stability) for TBP in CNOT1-LKO mice. It has already been shown that the CNOT complex has a role in mRNA splicing. For example, the CNOT7 variant 2 regulates the inclusion of CD44 variable exons [92]. The CD44 antigen is a non-kinase transmembrane glycoprotein involved in cell–cell interactions, cell adhesion and migration (metastasis). Moreover, CNOT3 level is inversely proportional to PRPF31, which is an RNA binding protein and splicing factor, which forms part of the spliceosome. CNOT3 binds to the PRPF31 promoter and negatively regulates the expression of PRPF31 [95]. Furthermore, we have shown that TBP mRNA stability increases in the absence of CNOT1, which probably explains its elevated protein expression in CNOT1-depleted cells.

Transcription-associated R-loop formation in CNOT1-depleted cells was confirmed using slot-blot assays and immunoblotting with the S9.6 antibody. The S9.6 signal was diminished via pre-treatment of gDNA with RNase H1 enzyme, confirming that the signal was derived from RNA. In addition, the immunofluorescence detection of nuclear S9.6 was consistent with the results obtained from the slot-blot assay. The transfection of RNase H1 pcDNA for 24 h decreased the S9.6 intensity in both control and CNOT1-depleted cells. The ability of the endogenous RNase H1 enzyme to remove R-loops from gDNA, determined in the slot-blot assay, was somewhat more efficient than that of the transiently transfected recombinant RNase H1 in immunofluorescence detection of S9.6, probably due to an inefficiency of transfection in the latter case. Since the S9.6 antibody binds to double-stranded RNA (dsRNA) as well as RNA–DNA hybrids and can give rise to nonspecific signals, in a separate experiment, we measured the R-Loop formation in U2OS cells stably expressing the GFP-tagged, catalytically active and inactive RNase H1 proteins. CNOT1 depletion significantly elevated the GFP signal in dRNH1 compared to its control conditions. This upregulation was at an even higher level in GFP-dRNH1 U2OS cells compared to GFP-wtRNH1 U2OS cells, confirming that RNaseH1 exhibits specific activity toward RNA–DNA hybrid removal. Double siRNA-mediated depletion of CNOT7 and 8 increased the GFP signal in dRNH1 U2OS cells compared to control. However, the level of S9.6 signal was significantly higher in CNOT7-KO and CNOT8-KO HEK293T cells compared to the control cells. This may also reflect the inefficiency of siRNA-mediated depletion compared to CRISPR-mediated knock out.

We have also shown that increased transcription-dependent R-loop formation can be a source of slowed replication forks in CNOT1-depleted cells. In these experiments, transfection with recombinant RNase H1 did not have a significant effect on replication initiation, although it rescued replication fork progress in CNOT1-depleted cells to a limited extent, but not in control cells.

Several studies have demonstrated a role of the deadenylase activity of the CNOT complex in the DNA damage response in *Saccharomyces cerevisiae* [51,96]. In mammalian cells, it has been shown that CNOT6 depletion resulted in a remarkable resistance to cisplatin-mediated apoptosis following the induction of Chk2 T68 phosphorylation [52]. Additionally, the depletion of CNOT6 increased the stability of mRNA transcripts of mismatch repair (MMR) genes, leading to the increased expression of MMR proteins and a decrease in the mutation frequency in MMR-proficient cells [97]. In addition, the deficiency in transcription-coupled repair (TC-NER), a sub-pathway of NER, is associated with mutations in the Ccr4–Not complex in yeast, suggesting additional functions in DNA repair during transcription elongation [49], which is consistent with the data presented here. The induction of DNA damage in CNOT1-depleted cells was confirmed via increased Olive tail moment, micronuclei formation, 53BP1 foci formation (in G1 positive cells) and increased chromosomal aberrations, such as gaps/breaks and radials.

Persistent stalled DNA replication forks ultimately collapse into DSBs with a subsequent activation of the ATM/Chk2 signalling pathway. CNOT1 reduction in HeLa cells led to increased phosphorylation of ATM and its substates Chk2, H2AX and KAP1. In contrast, the ATR/Chk1 repair pathway was downregulated in cells lacking CNOT1. We have also shown that ATR, which acts as a safeguard to detect replication blocks, associates with CNOT1 and Tab182, possibly indicating an interaction with the intact CNOT complex.

In an attempt to better understand the effects of CNOT1 depletion on cellular pathways, an RNA-Seq analysis was carried out on HeLa cells treated with control or CNOT1 siRNA for 72 h. As expected, the reduction in CNOT1 expression affected multiple pathways, most notably causing an upregulation of genes involved in cytokine signalling and the cellular response to cytokines. Significantly, there was a marked upregulation of genes involved in the regulation of the ERK1 and ERK2 cascade. This aspect of CNOT1 depletion was investigated in more detail. The hyperactivation of the ERK1/2 signalling pathway and cyclin E upregulation in CNOT1-depleted cells led to the accumulation of cells at the border of the G1/S phase, where DNA origin firing is licensed. These data were consistent with FACS analysis of cell cycle profiles of control and CNOT1-depleted cells, which showed arrest of CNOT1-depleted cells in G1/S and an inability to progress through the cell cycle. Previous studies had proposed a novel role for Ccr4–Not in the regulation of mRNAs involved in cell cycle progression in yeast [88]. These investigators showed that the interaction between Spt6, a histone chaperone, and RNA PoII promotes the recruitment of the Ccr4–Not complex at the site of transcription to facilitate the degradation of mRNAs required for cell cycle progression. This recruitment possibly occurs during the G1 phase as the CNOT complex is concentrated in the nucleus in early G1 cells; however, by the time the cells enter the S phase, much of it becomes cytoplasmic [98]. Consistently with this, we have shown that the depletion of CNOT1 significantly reduced the ability of CNOT1-depleted cells to form colonies 14 days after the plating cells (in the absence of DNA-damaging agents). We have also shown that, at later times, CNOT1-depleted HeLa cells appeared to senesce (increased senescence-associated β-galactosidase activity), whereas control cells did not. No activation of caspase-3 or degradation of PARP1, which are normally taken as diagnostic markers of apoptosis, was observed in CNOT1-depleted HeLa cells (Appendix A). However, the expression of caspase types 1 and 4 were both upregulated in CNOT1-depleted HeLa cells (data collected from the RNA-Seq). It had previously shown that a sub-G1 apoptotic fraction was increased in CNOT1-depleted HeLa cells [55]. In this study, it was further shown that CNOT1 depletion increased CHOP mRNA levels and activated caspase-4, which is associated with ER stress-induced apoptosis [55].

Since the CNOT complex is a multi-subunit and multi-functional protein complex, we cannot confirm whether the phenotypes we observed in CNOT1-depleted HeLa cells are the direct effect of CNOT1 depletion or a consequence of the depletion/inactivation of the other CNOT subunits and their consequent enzymatic activities (deadenylation and ubiquitylation). However, these observations provide novel insights into the ways in which the inactivation of the CNOT complex can lead to DNA damage and genomic instability. However, it is notable that the loss of CNOT7 or CNOT8 appears to result in genome instability as evidenced by increased micronuclei formation and DNA damage foci (in the absence of exogenous damage) compared to controls.

One of the mechanisms proposed to cause genomic instability, possibly as a forerunner of cancer, is replication stress. There are many sources of replication stress; to date, however, relatively little attention has been paid to transcription as a major cause of endogenous replication stress and DNA damage in the context of activity of the CNOT complex. Here, we propose a model introducing a novel role for CNOT complex in regulation of genome stability via transcription activity (Figure 8). Cancer is a disease of genomic instability, characterized by high mutation rates and genomic rearrangements that ultimately drive aggressiveness and resistance to therapy. As CNOT1 is essential for CNOT function, it is not surprising that CNOT1 knock-out mice die during development [99]. However, SNPs in the CNOT1 gene have been reported in B-cell paediatric ALL [100] and missense mutations have been frequently reported (376 out of 481 total mutations) in colorectal, melanoma, uterine and many other cancers (cbioportal project). The low expression of CNOT1 increases the overall rate of survival (generated using the Kaplan–Meier method) in gastric cancer and osteosarcoma [100]. However, the level of CNOT1 expression (high or low) makes no significant difference in time in prostate and lung cancers [101,102]. A careful consideration of the impact of mutations/deletions of all CNOT complex genes, in relation to genomic instability, seems likely to yield insights into a possible contribution to cancer development in the future.

## Figures and Tables

**Figure 1 cells-12-01868-f001:**
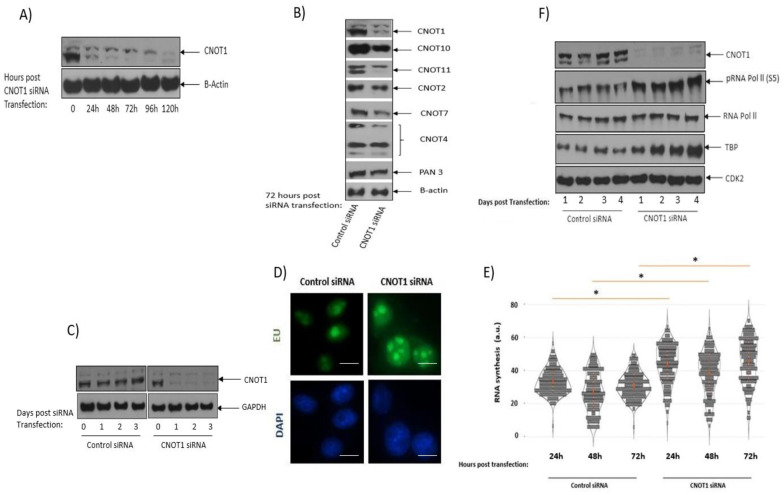
CNOT1 depletion increases global transcription in HeLa and MCF7 cells. (**A**) Representative immunoblot shows CNOT1 depletion at different time points (0, 24 h, 48 h, 72 h, 96 h and 120 h) after transfection of HeLa cells with CNOT1 siRNA. (**B**) siRNA-transfected HeLa cells were lysed, and the lysates assessed by immunoblotting with the indicated antibodies against different CNOT subunits and Pan 3. (**C**) Representative immunoblot confirms depletion of CNOT1 via siRNA up to 72 h. (**D**) Representative IF images show the labelling of global nascent RNA using 5-ethynyluridine (EU) incorporation assay in control and CNOT1-depleted HeLa cells. The global transcription activity was measured after EU incorporation (1 h) shown after 24, 48, and 72 h in (**E**). To quantify EU incorporation, Hoechst was used to stain the nucleus of cells, generating a nuclear mask. Adobe Photoshop was used to adjust the levels of all panels equally. Quantification of EU signal per nucleus was performed using RStudio statistical software and the results shown (*n* = 3 independent experiments, 100 cells counted. Stats: Mann–Whitney Wilcoxon Rank Sum). (**F**) Representative Western blot showing the expression level of TBP and the phosphorylation of RNA Polymerase II C-terminal domain (S5) in each experimental condition. CDK2 was used as a loading control. (**G**) Representative Western blot showing the expression level of CNOT subunits in the presence or absence of CNOT1 in MCF7 TR/ Control and CNOT1 KD cells. (**H**) Representative Western blot confirms the CNOT1 (flag-tagged) pcDNA3 transfection in MCF7 TR/CNOT1 KD cells (Transfection efficiency was generally in the range 20–25%). (**I**,**J**) Representative images show transfection efficiency (22%). Images were collected using an EVOS fluorescence inverted digital microscope. MCF7 cells were fixed, permeabilized, and labelled with anti-FLAG antibody (green). Percentage transfection efficiency: (Fluorescent cells/Total number of cells) × 100. Bar 400 µm. (**K**,**L**) Transcription elongation was measured after EU incorporation (1 h) 3-day post +/− 2 μg/mL DOX treatment in MCF-7 TR control and MCF7 TR CNOT1KD cells (Representative cells are shown in (**H**)). To quantify EU incorporation, Hoechst was used to stain the nucleus of cells, generating a nuclear mask. Adobe Photoshop was used to adjust the levels of all panels equally. Quantification of EU signal per nucleus was performed using GraphPad Prism statistical software; (*n* = 3 independent experiments; >100 cells analysed per repeat; Stats: unpaired *t*-test, * *p* < 0.05). (**M**) Representative Western blot showing the expression level of TBP and the phosphorylation of RNA Polymerase II C-terminal domain (S5) in MCF-7 TR CNOT1KD cells up to 5 days post 2 μg/mL DOX induction. Scale bar in (**C**,**H**) = 50 µm. a.u, arbitrary units.

**Figure 2 cells-12-01868-f002:**
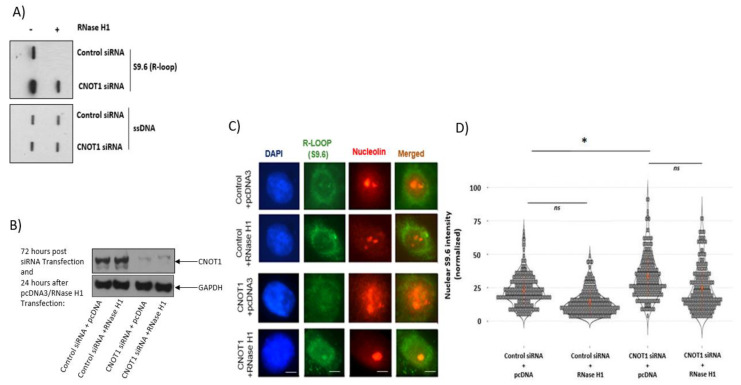
Depletion of CNOT1 leads to increased R-loop formation. (**A**) Detection of RNA–DNA hybrids using slot blot analysis and S9.6 antibody on gDNA isolated from HeLa cells 72 h post CNOT1 depletion. A ssDNA antibody was used as a loading control (lower panel). (**B**) Representative Western blot confirms the efficiency of CNOT1 depletion up to 96 h. (**C**) Co-immunostaining of HeLa cells with S9.6 (green) and nucleolin (red) antibodies 72 h post siRNA transfection and 24 h after recombinant RNase H1 transfection. The nuclear DNA was stained with DAPI. (**D**) Quantification of S9.6 antibody signal per nucleus after subtraction of nucleolar staining using Image J. Statistical analysis was performed using RStudio statistical software. (*n* = 3 independent experiments, 100 cells counted per experiment. Stats: Mann–Whitney Wilcoxon Rank Sum). (**E**) Co-immunostaining of MCF-7 TRCNOT1KD cells with S9.6 (green) and nucleolin (red) antibodies 72 h post +/− 2 μg/mL DOX treatment. (**F**) Quantification of S9.6 antibody signal per nucleus after subtraction of nucleolar staining using Image J. Statistical analysis was performed using GraphPad Prism statistical software 9.5.1; (*n* = 3 independent experiments; >100 cells analysed per repeat; Stats: unpaired *t*-test, * *p* < 0.05,). (**G**,**H**) Representative images show percentage transfection efficiency (21%). Images were collected using an EVOS fluorescence inverted digital microscope. MCF7 Cells were fixed, permeabilized, and labelled with anti-FLAG antibody (green). Percentage transfection efficiency: (fluorescent cells/total number of cells) × 100. Bar 400 µm. (**I**,**J**) Representative IF images and bar graph showing the GFP signal obtained from control siRNA, CNOT1 siRNA, CNOT4 siRNA and double CNOT7/8 knock down and in dead-RNH1 U2OS cells. The intensity of GFP signal was equalized across different samples using Image J. Statistical analysis was performed using excel; (*n* = 3 independent experiments; >100 cells analysed per repeat; Stats: unpaired *t*-test, * *p* < 0.05,). Scale bar = 50 µm. AU, arbitrary units. (**K**,**L**) Representative Western blot showing the expression of CNOT 1, 7/8 and CNOT4 subunits 3 days post siRNA transfection in In GFP dRNH1 U2OS cells.

**Figure 3 cells-12-01868-f003:**
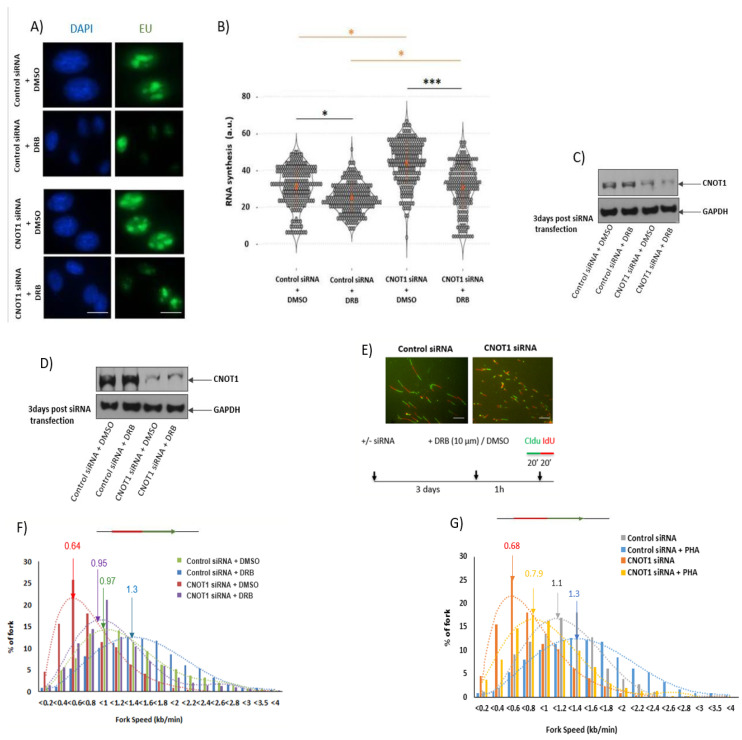
Depletion of CNOT1 induces replication stress through ongoing transcription. (**A**) Inhibition of RNA synthesis with DRB in CNOT1-depleted HeLa cells. Cells were treated with CNOT1 or control siRNA. A total of 72 h post siRNA transfection cells were exposed to 100 µM DRB or solvent DMSO for 100 min during EU labelling. (**B**) Quantification of EU signal per nucleus treated with DRB was performed using RStudio statistical software and shown (*n* = 3 independent experiments Stats: Mann–Whitney Wilcoxon Rank Sum). Statistical analyses were performed using a two-tailed and unpaired Student’s *t* test, * *p* < 0.05, *** *p* < 0.001 (**C**,**D**) Representative Western blot confirms the efficiency of CNOT1 depletion on day 3 following treatment with DMSO or DRB for IF and fibre assay, respectively. (**E**) Experimental scheme of dual labelling of DNA fibres in HeLa cell lines. (**F**) Distribution of replication fork speeds 72 h after CNOT1 depletion; HeLa cells were treated with DRB (450 replication forks were examined from three independent experiments). (**G**) Distribution of replication fork speeds 72 h after CNOT1 depletion in HeLa cells treated with 10 µM PHA- 767491 for 1 h (450 replication forks were examined from three independent experiments). The fibre lengths obtained in Image J were converted into micrometres using the scale bars on the microscope. (**H**,**I**) Representative images show percentage transfection efficiency (approximately 25%). Images were collected using an EVOS fluorescence inverted digital microscope. MCF7 Cells were fixed, permeabilized, and labelled with anti-FLAG antibody (green). Percentage transfection efficiency: (fluorescent cells/total number of cells) × 100. Bar 400 µm. (**J**) Representative image showing fields of DNA fibres in MCF-7 TRCNOT1KD cells. The DNA fibres pictured were labelled sequentially with CldU and IdU for 20 min each. Scale bar = 50 µm. (**K**) Distribution of replication fork speeds 72 h post +/− 2 μg/mL DOX treatment and 2 days post CNOT1 pcDNA3 transfection. A total of 250 replication forks were examined from three independent experiments. The fibre lengths obtained in Image J were converted into micrometres using the scale bars on the microscope.

**Figure 4 cells-12-01868-f004:**
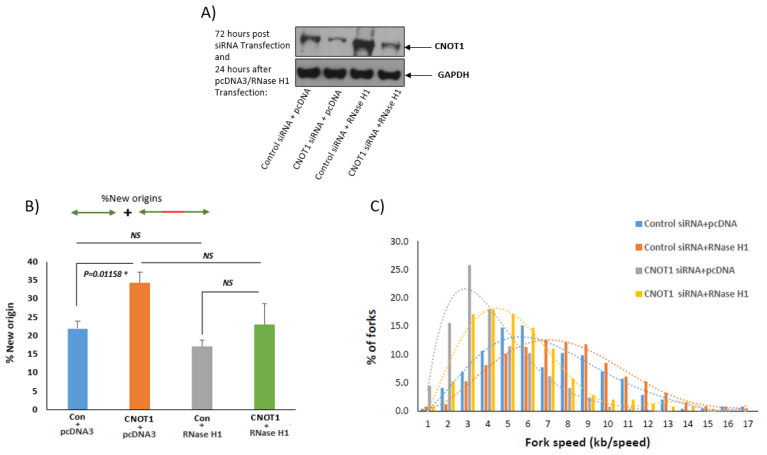
CNOT1-depleted HeLa cells have increased R-loop accumulation. (**A**) Representative Western blot confirms the efficiency of CNOT1 depletion up to 96 h. (**B**) Representative bar graph shows the percentage of new origins during DNA fibre labelling 72 h post CNOT1 depletion and 24 h post transfection with pCMV6- AC-RNase H1-GFP or pcDNA 3.1 (450 replication forks were analysed from three independent experiments). Statistical analyses were performed using a two-tailed and unpaired Student’s *t* test, Error bars represent SD. (**C**) Distribution of replication fork speeds 24 h post transfection with pCMV6-AC-RNase H1-GFP or pcDNA 3.1 following 72 h CNOT1 depletion (450 replication forks were analysed from three independent experiments. The lengths obtained in Image J were converted into micrometres using the scale bars on the microscope.

**Figure 5 cells-12-01868-f005:**
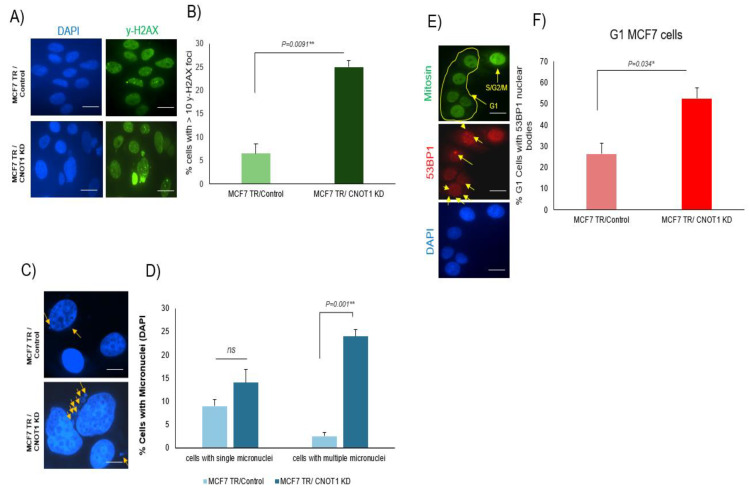
Increased genome instability in CNOT1-depleted cells. (**A**,**B**) Quantification of cells with >10 γ-H2AX foci in MCF-7 cells 3 days post +/− 2 μg/mL DOX induction. (**C**,**D**,**H**,**I**) Quantification of siRNA treated HeLa cells and MCF TRCNOT1KD cells with micronuclei classified in two categories (single micronuclei per cell and multiple micronuclei per cell). Scale bars, 5 μm. (*n* = 3 independent experiments; >100 cells counted per repeat, mean ± SD, * *p* < 0.05, ** *p* < 0.01 ). In (**C**,**H**) arrows indicate micronuclei. (**E**,**F**,**J**,**K**) Quantification of CNOT1 siRNA-treated HeLa and MCF TRCNOT1KD cells with 53BP1 nuclear bodies (indicated by arrows in central panel) in G1 positive control as judged by co-staining with either Cyclin A (**J**) or Mitosin (**E**) (in E, upper pane, and J, central panel, arrows indicate cells in a particular indicated phase of the cell cycle). (**L**) The level of DSBs, assessed using neutral comet assay, was performed 120 h post siRNA transfection in control and CNOT1-depleted HeLa cells. Dot plot of Olive tail moments was performed using RStudio statistical software (*n* = 3 independent experiments; >100 cells analysed per repeat, mean ± SD, ** *p* < 0.01 and). (**M**,**N**) Quantification of chromatid gaps and breaks per 100 metaphase spreads in control and CNOT1-depleted HeLa cells (* *p* < 0.05 and ** *p* < 0.005; *n* = 3). Scale bar = 50 µm. (**G**) Representative Western blot confirms the efficiency of CNOT1 depletion on day 3.

**Figure 6 cells-12-01868-f006:**
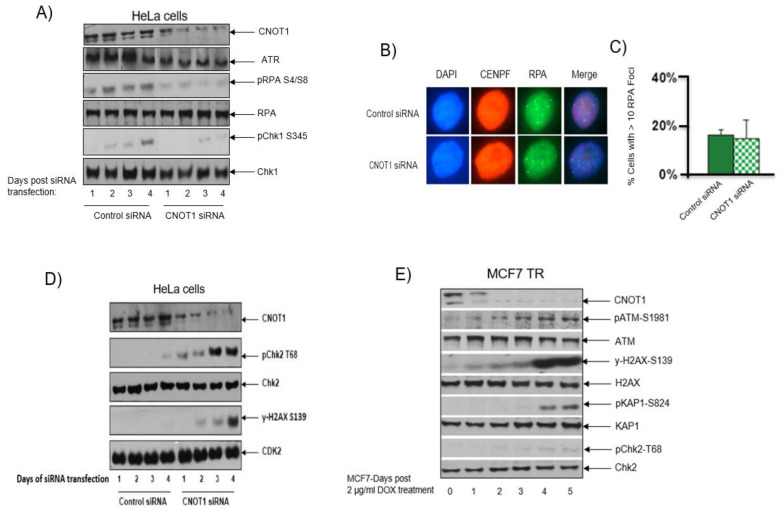
CNOT1 depletion leads to down regulation of ATR/Chk1 and activation of ATM/Chk2 repair pathways. (**A**) Western blot showing the expression and phosphorylation of ATR, ATRIP, Chk1, RPA in control and CNOT1 depleted HeLa cells up to 4 days post siRNA transfection. Actin was used as a loading control. (**B**,**C**) Quantification of cells with >10RPA foci in both +/− CENPF control and CNOT1 depleted HeLa cells up to 3 days post siRNA transfect ion. (**D**,**E**) Western blot showing the expression level of total ATM, Chk2, H2AX, KAP1 in HeLa and MCF7 cells up to 4 days siRNA transfection and 3 days post DOX treatment, respectively.

**Figure 7 cells-12-01868-f007:**
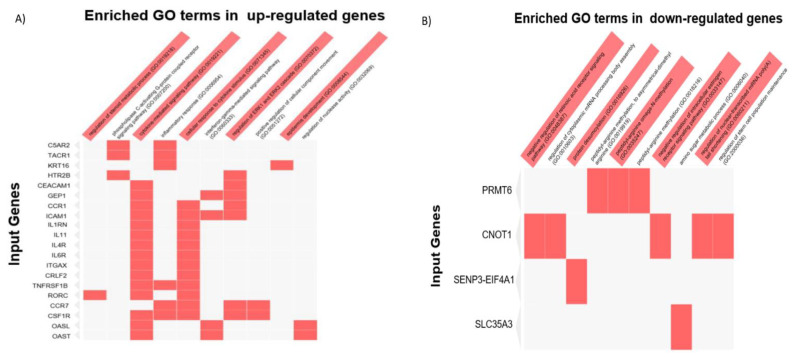
DNA damage-induced cell cycle arrest and senescence in CNOT1-depleted HeLa cells. (**A**,**B**) Up- and downregulated genes 72 h post CNOT1 siRNA treatment in HeLa cells. (**C**–**E**) Representative immunoblots showing the comparative protein expression between control and CNOT1-depleted HeLa cells. GAPDH and CDK2 were used as loading controls. (**F**,**G**) HeLa cells were transfected with either Control or CNOT1 siRNA and stained with PI and the cell cycle profiles analysed by flow cytometry at different time-points post-transfection. Representative profiles of the cells 72 h post transfection are shown. (**H**) HeLa cells were plated at appropriate concentrations 72 h post CNOT1 and control siRNA transfection. At day 14, large colonies were stained with crystal violet and counted. (**I**) Bar graph shows the plating efficiency of the cells at day 14 after staining with crystal violet (*n* = 3 independent experiments). Statistical analyses were performed using a two-tailed and unpaired Student’s *t* test, ***, *p* < 0.001. Error bars represent SD. (**J**) Representative images showing the development of SA-β-Gal staining (positive senescent cells) in control HeLa cells (upper panel) and CNOT1-depleted HeLa cells (lower panel) between day 3 (left panel) and day 7 (right panel) post siRNA transfection. The blue color represents senescent cells. Scale bar = 50 μm. (**K**) Bar graphs showing the percentage of SA-β-Gal positive cells in control and CNOT1-depleted HeLa cells 3-, 5- and 7-days post siRNA depletion (*n* = 3 independent experiments). Statistical analyses were performed using a two-tailed and unpaired Student’s *t* test, * *p* < 0.05. Error bars represent SD.

**Figure 8 cells-12-01868-f008:**
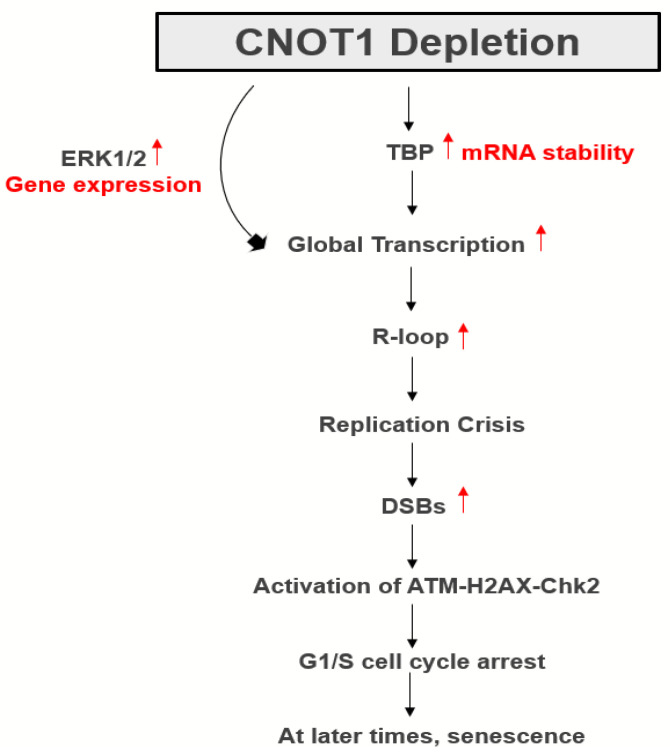
Proposed model showing how disruption of the CCR4–Not complex contributes to transcription-mediated genome instability via RNA–DNA hybrids’ formation. Red arrow indicates an increase.

**Table 1 cells-12-01868-t001:** siRNAs used in this study.

Target	siRNA	Sense Sequence	Antisense Sequence	Supplier
CNOT1	SMARTpool	CUAUAAAGAGGGAAGAGA CCAGAAACUUUGGCGACAA GGCCAAAUUGUCUCGAAUA CAAGUUAGCACUAUGGUAA	UCUCGUUCCCUCUUUAUAG UUGUCGCCAA AGUUUCUGG UAUUCGAGAC AAUUUGGCC UUACCAUAGU GCUAACUUG	Dharmacon
CNOT7	SMARTpool	CAGCUAGGACUGACAUUUA GGAGAAUUCAGGAGCAAUG UCAUAGCGGUUACGACUUU GUUAGAGCUGGAACGGAUA	UCUCGUUCCCUCUUUAUAG UUGUCGCCAA AGUUUCUGG UAUUCGAGAC AAUUUGGCC UUACCAUAGU GCUAACUUG	Dharmacon
CNOT8	SMARTpool	UUUCGUAGUUCCAUAGAU GAGAAUAGCCAGGUUAUCU AAUAUCAGCUUCUGCGGUG CCAUAGAUCUCCUUGCUAA	AUCUAUGGAACUACGAAA AGAUAACCUGGCUAUUCUC CACCGCAGAAGCUGAUAUU UUAGCAAGGAGAUCUAUGG	Dharmacon
lacZ 198-non silencing control	Custom	CGUACGCGGAAUAC UCGAdTdT	AhAhUCGAAG UAUUCCGCGUACG	Dharmacon

**Table 2 cells-12-01868-t002:** Antibodies used in Western blotting studies.

Primary Antibody	Species	Application	Dilution	Supplier
CNOT1	Rabbit	WB/IP	1:500	Proteintech
CNOT2	Rabbit	WB	1:500	Proteintech
CNOT 3	Rabbit	WB	1:500	Proteintech
CNOT4	Rabbit	WB	1:500	SantaCruz
CNOT4	Rabbit	WB	1:500	Proteintech
CNOT6	Rabbit	WB	1:500	SantaCruz
CNOT7	Rabbit	WB	1:500	SantaCruz
CNOT8	Rabbit	WB	1:500	Proteintech
CNOT9	Rabbit	WB	1:500	Proteintech
CNOT10	Rabbit	WB	1:500	SantaCruz
CNOT11	Rabbit	WB	1:500	Proteintech
Pan3	Rabbit	WB	1:500	SantaCruz
53BP1	Rabbit	IF/WB	1:1000	Novus
p53 (DO1)	Mouse	WB	1:2000	Gift from D. Lane
H2AX	Rabbit	WB	1:1000	Millipore
γH2AX (S139)	Mouse	WB/IF	1:1000	Millipore
Chk2	Rabbit	WB	1:1000	Gift from S. Elledge
pChk2 (T68)	Rabbit	WB	1:1000	Cell Signalling
IdU	Mouse	DNA fibres	1:500	Becton Dickinson
CldU	Rat	DNA fibres	1:750	AbD SeroTec
ERK1/2	Rabbit	WB/IF	1:500	Cell Signalling
pERK1/2	Rabbit	WB/IF	1:500	Cell Signalling
p21Cip1	Rabbit	WB	1:500	Abcam
TBP	Rabbit	WB	1:1000	SantaCruz
RNA Po II C-terminal domain (CTD)	Rabbit	WB	1:1000	SantaCruz
β-Actin	Mouse	WB	1:1000	Sigma-Aldrich
GAPDH	Mouse	WB	1:500	SantaCruz
R-loop (S9.6)	Mouse	WB	1:1000	Gift from E. Peterman
ssDNA	Mouse	WB	1:5000	Millipore
Cyclin A	Rabbit	WB/IF	1:500	SantaCruz
Cyclin B	Rabbit	WB	1:500	SantaCruz
Cyclin E	Rabbit	WB	1:500	Abcam
Cyclin D1	Mouse	WB	1:500	SantaCruz
pHistone H3 (Ser- 10)	Rabbit	WB	1:100	Cell signalling
Mitosin	Mouse	WB/IF	1:500	BDBiosciences
ATM	Mouse	WB	1:1000	Gift from P. Byrd
pATM (S1981)	Rabbit	WB	1:500	R & D Systems
KAP1	Rabbit	WB	1:1000	Bethyl Laboratories
pKAP1 (S824)	Rabbit	WB	1:1000	Bethyl Laboratories
MCM2	Rabbit	WB/IP	1:500	SantaCruz
MCM7	Rabbit	WB/IP	1:500	SantaCruz
Tab182	Rabbit	WB/IP	1:3000	In house
TOPBP1	Rabbit	WB	1:1000	Bethyl
ATRIP	Rabbit	WB, IP	1:500	Abcam
PARP1	Mouse	WB	1:1000	SantaCruz

## Data Availability

All data is available on request.

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
