# Peer review of "Disruption of the Mammalian Ccr4–Not Complex Contributes to Transcription-Mediated Genome Instability"

_cells, 2023, doi:10.3390/cells12141868_

Round 1
Reviewer 1 Report
The manuscript by Hagkarim et al. documents a new function of the multifunctional complex, CCR4-NOT, in maintaining genomic stability. The authors show that depletion of CNOT1, the scaffold component of the CCR4-NOT complex, increases global transcription and subsequent formation of the R-loop intermediates and replicative stress. Transcriptome analysis in HeLa cells revealed that CNOT1 knockdown results in hyperactivation of the ERK signalling something that associates with cell cycle arrest and senescence. These are interesting observations although a causal link between CNOT1-dependent ERK activation and cell cycle arrest was not reported. Overall, this study suggests a potential new role of Cnot in genome stability that could be of interest to Cells readership. However, a major concern of the reviewers, which needs to be addressed prior to publication, is that all the experiments presented in this manuscript have been performed with a pool of four siRNAs, which are notorious for their off-target effects (even when on-target plus reagents are being used). Therefore, the authors should repeat all key experiments using at least three independent siRNAs and ideally they should also perform rescue experiments with CNOT1 cDNA.
Major comments:
1. As mentioned above, the authors should use multiple independent siRNAs and perform rescue experiments of all the key phenotypes observed. These include the increase of global transcription, R-loop formation, genome instability and ERK signaling following CNOT1 depletion. These experiments are essential for ensuring that the conclusions of the study are accurate.
2. The manuscript does not explain the role of other components of the CCR4-NOT complex in the observed phenotypes. Especially considering that CNOT1 depletion results in destabilization of other CCR4-NOT complex components, the experiments to discriminate the function of CNOT1 and other members of the CCR4-NOT complex should be tested. For example, the enhanced transcription and R-loop formation could be evaluated by knocking down other components of the CCR4-NOT complex to explain if only CNOT1 or all the CCR4-NOT complex proteins contribute to the observed phenotypes. This is particularly important for disentangling between CNOT1 and the different enzymatic activities of the CCR4-NOT complex (deadenylation and ubiquitination). One is left wondering if the observed phenotypes are a direct consequence of CNOT1 loss or due to subsequent disruption of the complex and its associated enzymatic activities. Thus, the authors should evaluate the increase in transcription and R-loop formation by knocking down proteins with deadenylase activity (CNOT7 and CNOT8) and E3 ligase activity (CNOT4). Additionally, the increased transcription and R-loop formation following CNOT1 depletion should be further evaluated by rescuing with CNOT1 deletion mutants that fail to interact with deadenylating components (Raisch et al., Nat Commun 10, 3173 (2019)).
Minor points:
1. Association of TBP with increased transcription, R-loop formation, and replicative stress has been reported and should be cited (Kotsantis et al., Nat Commun 7, 13087 (2016)). It has been reported that increased transcription, R-loop formation, and replicative stress were dependent on the increase in TBP protein levels, which was also observed in this study (Figure 1E). The authors should also evaluate or discuss how CNOT1 depletion results in up-regulation of TBP. Does the RNA seq analysis show differential expression of genes that correlate with TBP upregulation (such as MYC, RAS, CHEK1, and CHEK2)? This would help to explain the connection between CNOT1 depletion and some of the observed phenotypes such as increased transcription, R-loop formation, replicative stress, and senescence.
2. The differential gene expression analysis data should be included as a supplementary excel file.
3. The tools used for differential gene expression analysis and the cutoff values used for foldchange and significance (FDR<0.05) should be indicated in the materials and methods section. The information in Figure 5 legends (lines 449 to 455) is already provided in the materials and methods section and can be excluded.
4. Ito et al., 2011 should also be referenced to support that Cnot1 depletion leads to destabilization of other proteins in the CCR4-NOT complex.
Author Response
Response to reviewers (in red).
We are most grateful to the reviewers for their very helpful suggestions. We have now modified the manuscript extensively in line with their criticisms. The major change has been to repeat many of the experiments using an MCF7 cell line with an inducible shRNA against CNOT1; furthermore, we have reversed the effects of depletion by transfecting back a plasmid encoding CNOT1 (Figures 1 [G, H, I], 2 [E] and 3 F]. In addition, we have looked at the effects of loss of two of the deadenylase subunits (CNOT7 and CNOT8) as suggested by the reviewers (Figure 2[G], Supplementary Figure S3, Supplementary Figure S6).
Reviewer 1
The manuscript by Hagkarim et al. documents a new function of the multifunctional complex, CCR4-NOT, in maintaining genomic stability. The authors show that depletion of CNOT1, the scaffold component of the CCR4-NOT complex, increases global transcription and subsequent formation of the R-loop intermediates and replicative stress. Transcriptome analysis in HeLa cells revealed that CNOT1 knockdown results in hyperactivation of the ERK signalling something that associates with cell cycle arrest and senescence. These are interesting observations although a causal link between CNOT1-dependent ERK activation and cell cycle arrest was not reported. Overall, this study suggests a potential new role of Cnot in genome stability that could be of interest to Cells readership. However, a major concern of the reviewers, which needs to be addressed prior to publication, is that all the experiments presented in this manuscript have been performed with a pool of four siRNAs, which are notorious for their off-target effects (even when on-target plus reagents are being used). Therefore, the authors should repeat all key experiments using at least three independent siRNAs and ideally they should also perform rescue experiments with CNOT1 cDNA.
We have now repeated many of the experiments using an alternative depletion of CNOT1 (the DOX inducible MCF7 cell line) (Figures 1 [F-I], 2 [D, E], 3 [E, F], 5 [A-D], 6[E]) and used transfected CNOT1 to reverse the effects of CNOT1 depletion (Figures 1 [I], 2[E], 3[F]).
Major comments:
- As mentioned above, the authors should use multiple independent siRNAs and perform rescue experiments of all the key phenotypes observed. These include the increase of global transcription, R-loop formation, genome instability and ERK signaling following CNOT1 depletion. These experiments are essential for ensuring that the conclusions of the study are accurate.
As mentioned, we have now repeated most of the experiments using an MCF7 cell line with an inducible shRNA allowing depletion of CNOT1 (Figures 1 [F-I], 2 [D, E], 3 [E, F], 5 [A-D], 6[E]) . In some cases, the knock down has been complemented with transfected CNOT1 DNA reversing the effects seen (Figures 1 (H &I), 2 (D & E), 3 (E & F) in the modified manuscript). Very sim ilar results were obtained with sioRNA depletion in HeLas and DOX inducible depletion in MCF7 cells. Also, transfection of CNOT1DNA reversed the effects of CNOT1 depletion to some effect. That we did not see total complementation was probably due to difficulties in transfecting such a large gene as CNOT1.
- The manuscript does not explain the role of other components of the CCR4-NOT complex in the observed phenotypes. Especially considering that CNOT1 depletion results in destabilization of other CCR4-NOT complex components, the experiments to discriminate the function of CNOT1 and other members of the CCR4-NOT complex should be tested. For example, the enhanced transcription and R-loop formation could be evaluated by knocking down other components of the CCR4-NOT complex to explain if only CNOT1 or all the CCR4-NOT complex proteins contribute to the observed phenotypes. This is particularly important for disentangling between CNOT1 and the different enzymatic activities of the CCR4-NOT complex (deadenylation and ubiquitination). One is left wondering if the observed phenotypes are a direct consequence of CNOT1 loss or due to subsequent disruption of the complex and its associated enzymatic activities. Thus, the authors should evaluate the increase in transcription and R-loop formation by knocking down proteins with deadenylase activity (CNOT7 and CNOT8) and E3 ligase activity (CNOT4). Additionally, the increased transcription and R-loop formation following CNOT1 depletion should be further evaluated by rescuing with CNOT1 deletion mutants that fail to interact with deadenylating components (Raisch et al., Nat Commun 10, 3173 (2019)).
We accept the reviewer’s criticism. In an ideal world we would have depleted each of the 4 deadenylase subunits and the E3 ligase separately and in various combinations and used the readouts we have described for the CNOT1 experiments. Lack of time and personnel has precluded this. However, we have looked at the effects of knocking out 2 of the deadenylase adaptor subunits (CNOTs 7 and 8) on some of the properties we have examined for CNOT1 depletion (Figure 2G, Supplementary Figures S3 and S6). in the modified manuscript). There is a limitation on further investigation of the effects of depletion of multiple deadenylases in that cells which have lost both CNOT 7 and CNOT8 together are not viable. We have not addressed loss of CNOT6 or 6L, as no knock-out cell line appears to be available and several siRNAs we have examined give very poor results.
We have also used SMARTPOOL siRNA to deplete CNOT4 in a limited number of experiments. For instance, depletion of CNOT4 showed no difference with control sample in GFP-dRNH1 U2OS cells or similar level of y-H2AX foci and micronuclei formation as seen in control HEK293 cells (data are not shown) (lines 552-4 and 772-6). Again, due to lack of time and personnel we have not pursued the CNOT4 depletion. Obviously, this would be interesting but is really a whole new project. We feel there is already a very large amount of data in our manuscript, and one has to stop at some point. We hope the editor and reviewers will understand.
Minor points:
- Association of TBP with increased transcription, R-loop formation, and replicative stress has been reported and should be cited (Kotsantis et al., Nat Commun 7, 13087 (2016)). It has been reported that increased transcription, R-loop formation, and replicative stress were dependent on the increase in TBP protein levels, which was also observed in this study (Figure 1E). The authors should also evaluate or discuss how CNOT1 depletion results in up-regulation of TBP. Does the RNA seq analysis show differential expression of genes that correlate with TBP upregulation (such as MYC, RAS, CHEK1, and CHEK2)? This would help to explain the connection between CNOT1 depletion and some of the observed phenotypes such as increased transcription, R-loop formation, replicative stress, and senescence.
These points have now been considered in the modified manuscript lines 460-466 and Figure S2. RNA seq analysis did not show TBP gene upregulation and this upregulation was only observed at the protein level. Previously, un-spliced TBP mRNA (pre-mRNA) level was shown to be lower in liver-specific CNOT1-KO mice (CNOT1-LKO) compared to control cells; in contrast, its relative mRNA expression level was significantly higher than in control cells (Takahashi, Suzuki et al. 2020). This shows either a quicker pre-mRNA processing (splicing) or longer mRNA lifespan (increased mRNA stability) for TBP in CNOT1-LKO mice. Here, we have shown that TBP mRNA stability was increased up to 6 hours post transcription inhibition using DRB treatment in MCF7 TR/ CNOT1 KD cells compared to control cells which could explain its upregulation at both mRNA and protein levels (Supplementary Figure S2).
- The differential gene expression analysis data should be included as a supplementary excel file.
This has now been included as requested (Supplementary Figure S8).
- The tools used for differential gene expression analysis and the cutoff values used for foldchange and significance (FDR<0.05) should be indicated in the materials and methods section. The information in Figure 5 legends (lines 449 to 455) is already provided in the materials and methods section and can be excluded.
As requested, these data have now been included in the M&M (lines 384-414). Also, the duplication in the legend to Figure 7 has now been deleted.
- Ito et al., 2011 should also be referenced to support that Cnot1 depletion leads to destabilization of other proteins in the CCR4-NOT complex.
This reference has now been included (The role of the CNOT1 subunit of the CCR4-NOT complex in mRNA deadenylation and cell viability. Ito K, Takahashi A, Morita M, Suzuki T, Yamamoto T. Protein Cell. 2011 Sep;2(9):755-63. doi: 10.1007/s13238-011-1092-4)
Reviewer 2
In this manuscript Hagkarim et al examine the effects of CNOT1 depletion on transcription and replication stress. They show that CNOT1 depletion causes increased transcription as judged by EU incorporation, pSer5 Pol II and RNA-seq, as well as destabilization of the CNOT complex. Increased transcription levels give rise to reduced replication speed, which may be due to DNA-RNA hybrids. CNOT1-depleted cells also show an increase in genomic instability. However, these claims have to be supported by appropriate control experiments, orthogonal methods and data analysis, as specified below.
Major remarks:
- Rescue experiments with siRNA resistant CNOT1 are missing throughout the manuscript to confirm that the observed phenotypes are indeed caused by CNOT1 depletion.
Experiments have now been repeated with the depletion of CNOT1 in the DOX inducible MCF7 cell line to compensate for possible off-target effects of the SMARTpool of siRNAs used in the original experiments. Depletion of CNOT1 has been complemented by transfection of additional protein in some experiments (see (Figures 1 (H &I), 2 (D & E), and 3 (E & F) in the modified manuscript). Addition of CNOT1 reversed much of the effect of CNOT1 depletion. That we did not see total complementation was probably due to difficulties in transfecting such a large gene as CNOT1.
- The S9.6 antibody is not specific for DNA-RNA hybrids as it also recognizes ssRNA and dsRNA (PMID: 33830170). Therefore, the authors should use an orthogonal method to show DNA-RNA hybrids in cells such as catalytically inactive RNAse H1 or the hybrid-binding domain (HBD) of RNAse H1. In Fig. 3A RNAse H1 treatment did not remove the S9.6 signal, suggesting that the signal comes from RNAs.
Although the S9.6 antibody is open to criticism, data from its use is generally accepted to show R-loop formation (for example, at least 21 references in 2022). However, we have now made use of the catalytically dead GFP-RNAse H1 as recommended by the reviewer in additional experiment. The two approaches gave comparable results (Figure 2G).
- 3B – it is impossible to say anything about R-loop levels based on these images. RNAse H1-transfected cells should be visualized based on GFP fluorescence.
RNAseH1-GFP staining has now been carried out as mentioned above (Figure 2G in the modified manuscript).
4) RNA-seq. Please include the following:
- total number, uniquely mapping, and multi mapping reads per sample
- MA plots for differential gene expression
- Browser tracks (or access to UCSC)
Description and discussion of the RNAseq data has been modified to cover the points raised by the reviewer. (Lines 384-414 and 1029-1064 Supplementary Figure S9) in the modified manuscript).
- Results in Fig.5 don’t tie in with the main message of the manuscript regarding genomic instability. It would make more sense to focus on upregulation of DNA repair genes, considering the increase in genomic instability.
It is not quite clear which parts of the old Figure 5 the reviewer means. We agree that some of these observations are slightly removed from the main focus of the manuscript; however, they demonstrate the effects of prolonged depletion of CNOT1 and are, therefore in our opinion, sufficiently interesting to report. They are now shown in figure 7 and discussion of them has been slightly modified.
The expression of some DDR proteins is shown in Figure 6, and they do not appear to be affected by CNOT1 depletion
Minor remarks:
- DRB is a CDK9 inhibitor.
This has been noted (line 593)
2) Microscopy images should have a higher resolution. They are currently blurry and distorted. Scale bars are missing.
The old microscopy images have now been replaced with a those at higher resolution.
3) Fig. 2C/D. Please provide high-resolution and enlarged pictures of the DNA fibre assay for all conditions used in C/D.
Pictures have now been included (Figure numbers have now been changed to Figures 3 C, D, E, F) as requested.
Reviewer 2 Report
In this manuscript Hagkarim et al examine the effects of CNOT1 depletion on transcription and replication stress. They show that CNOT1 depletion causes increased transcription as judged by EU incorporation, pSer5 Pol II and RNA-seq, as well as destabilization of the CNOT complex. Increased transcription levels give rise to reduced replication speed, which may be due to DNA-RNA hybrids. CNOT1-depleted cells also show an increase in genomic instability. However, these claims have to be supported by appropriate control experiments, orthogonal methods and data analysis, as specified below.
Major remarks:
1) Rescue experiments with siRNA-resistant CNOT1 are missing throughout the manuscript to confirm that the observed phenotypes are indeed caused by CNOT1 depletion.
2) The S9.6 antibody is not specific for DNA-RNA hybrids as it also recognizes ssRNA and dsRNA (PMID: 33830170). Therefore, the authors should use an orthogonal method to show DNA-RNA hybrids in cells such as catalytically inactive RNAse H1 or the hybrid-binding domain (HBD) of RNAse H1. In Fig. 3A RNAse H1 treatment did not remove the S9.6 signal, suggesting that the signal comes from RNAs.
3) Fig. 3B – it is impossible to say anything about R-loop levels based on these images. RNAse H1-transfected cells should be visualized based on GFP fluorescence.
4) RNA-seq. Please include the following:
1. total number, uniquely mapping, and multi mapping reads per sample
2. MA plots for differential gene expression
3. Browser tracks (or access to UCSC)
5) Results in Fig.5 don’t tie in with the main message of the manuscript regarding genomic instability. It would make more sense to focus on upregulation of DNA repair genes, considering the increase in genomic instability.
Minor remarks:
1) DRB is a CDK9 inhibitor.
2) Microscopy images should have a higher resolution. They are currently blurry and distorted. Scale bars are missing.
3) Fig. 2C/D. Please provide high-resolution and enlarged pictures of the DNA fibre assay for all conditions used in C/D.
Author Response

(The authors gave the same response as above.)

Round 2
Reviewer 1 Report
The authors have tried to address the previous reviewers’ comments and the current version of the manuscript is improved. We only have some minor comments:
· The information related to Figure 7D is difficult to comprehend due to the improper labeling of the figure (which is labeled identical to 7C). More information/labels in Figure 7D should be included to make it easier for the readers to understand.
· Extra spaces at lines 494, 555, 561, 676, 700, 776, 803, 890, 917, and 1045 need to be removed.
· The spelling of "RNA pol II" on line 1062, table 2, figure 1j should be corrected.
· The spelling of "depletion" on line 703 should be corrected. It is necessary to conduct a comprehensive spell check.
· The readership and reviewers would prefer to see the “data not shown” mentioned in lines 508, 510, 701, and 968.
· Image 5g should be made in high resolution since the micronuclei are not obvious (an image in greyscale rather than in blue, would make the micronuclei more obvious).
· The section starting on line 939 is not well integrated with the rest of the study and appears irrelevant. Overall, the authors with the help of the editors should put some effort to improve the written quality of the manuscript.
Author Response
Response to reviewers round 2
Reviewer 1
Comments and Suggestions for Authors
The authors have tried to address the previous reviewers’ comments and the current version of the manuscript is improved. We only have some minor comments:
- The information related to Figure 7D is difficult to comprehend due to the improper labeling of the figure (which is labeled identical to 7C). More information/labels in Figure 7D should be included to make it easier for the readers to understand.
Labelling has been corrected for Figure 7D. The rationale is explained in the text (page 23 central paragraph).
- Extra spaces at lines 494, 555, 561, 676, 700, 776, 803, 890, 917, and 1045 need to be removed.
Spaces removed.
- The spelling of "RNA pol II" on line 1062, table 2, figure 1j should be corrected.
This has been corrected
- The spelling of "depletion" on line 703 should be corrected. It is necessary to conduct a comprehensive spell check.
Corrected
- The readership and reviewers would prefer to see the “data not shown” mentioned in lines 508, 510, 701, and 968.
‘Data not shown’ has been removed. The effect of CNOT4 knock down (original lines 509-10) has now been included (Figure 2I). The CNOT4 data mentioned on original lines 700-1 has now been shown in supplementary figure S6C and D. Mention of the autophagy data in original lines 968 has been deleted.
- Image 5g should be made in high resolution since the micronuclei are not obvious (an image in greyscale rather than in blue, would make the micronuclei more obvious).
A new image has been included
- The section starting on line 939 is not well integrated with the rest of the study and appears irrelevant. Overall, the authors with the help of the editors should put some effort to improve the written quality of the manuscript.
This section has been introduced at the beginning of the previous section. The text has now been modified as appropriate to indicate this (lines 1120 onwards).

Reviewer 2 Report
1) The rescue experiments should be performed in HeLa as well and in Fig. 1H the authors should stain for FLAG-CNOT1 transfected cells. The cells shown in the figure do not differ from CNOT1 KD cells and it could be that they were not transfected with CNOT1. Transfection efficiency should be determined as in all experiments CNOT1 re-expression does not rescue the phenotypes.
2) Fig. 2B/C: siCNOT1 data is missing; Fig. 2C is truncated and the sample order should be concordant with 2B
3) Fig. 2D: rescue images are missing
4) Fig. 2G: images are missing
5) Fig. 2B+D+G: FLAG-CNOT1 staining should be shown to visualize transfected cells
6) Fig. 2G: blot is truncated and should be shown as a separate panel (+figure legend)
7) All data should be shown or statements including ‘data not shown’ should be removed.
8) Problems with syntax: DRB is a relatively selective inhibitor of Cdk9, the kinase of the positive transcription elongation factor b (P-TEFb) required for processive transcription elongation by RNA polymerase II (Jonkers and Lis 2015) was also used, (Chao and Price 2001).
9) Fig. 3: Reduced DNA fork speed was not rescued with re-expression of CNOT.
10) English editing is required.
Author Response
Response to reviewers round 2
Reviewer 2
- The rescue experiments should be performed in HeLa as well and in Fig. 1H the authors should stain for FLAG-CNOT1 transfected cells. The cells shown in the figure do not differ from CNOT1 KD cells and it could be that they were not transfected with CNOT1. Transfection efficiency should be determined as in all experiments CNOT1 re-expression does not rescue the phenotypes.
We have now included photomicrographs of cells transfected with CNOT1-Flag (Figure 1J, 2H, 3I) and efficiency of transfection is given in the appropriate figure legends. The picture shown in Figure 1H has now been replaced with a more appropriate figure showing reduction of EU staining (new figure 1K). We acknowledge that the complementation of the MCF7 cells does not return the properties under examination to wt levels, but this is almost certainly due to difficulties in transfection because CNOT1 is such a large gene (2376 amino acids, 7128bp and a protein molecular weight of approximately 265K). Considering that transfection efficiency was 20-30% the values obtained for complemented cells are as would be expected.
The reviewer requests that complementation now be repeated in HeLa cells. We feel that this is unreasonable-the original request in the reviewer’s first report was for rescue experiments to be carried out. We did this using the MCF7 cells with inducible knock-down and complementing with transfected CNOT1-this had the effect of partially reversing the effects of CNOT1 depletion in several experiments (Figures 1, 2, 3). Although reversal was not complete, it was appreciable, particularly in view of the size of the gene. We have shown similar effects of CNOT1 depletion in 2 distinct cell lines (MCF7 and HeLa) using 2 different ‘depletion systems’ (transfected siRNA and inducible knock down). Furthermore, we have complemented the depletion for several of the experiments-we consider that this is sufficient to show that the effects seen are not the result of ‘off-target effects’ and are genuine.
- 2B/C: siCNOT1 data is missing; Fig. 2C is truncated and the sample order should be concordant with 2B
siRNA knock down has now been included (new Figure 2B). Figure 2C has been fixed and the order of 2C and 2D synchronised.
- 2D: rescue images are missing
Rescue images now included (new Figure 2 G/H).
- 2G: images are missing
These have now been included in Figure 2.
- 2B+D+G: FLAG-CNOT1 staining should be shown to visualize transfected cells
Flag CNOT1 is shown in Figure 2G/H (Original figures D and G were not complementation figures).
- 2G: blot is truncated and should be shown as a separate panel (+figure legend)
Truncation has now been remedied and blot fixed.
- All data should be shown or statements including ‘data not shown’ should be removed.
‘Data not shown’ has been removed. The effect of CNOT4 knock down (original lines 509-10) has now been included (Figure 2I). The CNOT4 data mentioned on original lines 700-1 has now been shown in supplementary figure S6C and D. Mention of the autophagy data in original lines 968 has been deleted.
- Problems with syntax: DRB is a relatively selective inhibitor of Cdk9, the kinase of the positive transcription elongation factor b (P-TEFb) required for processive transcription elongation by RNA polymerase II (Jonkers and Lis 2015) was also used, (Chao and Price 2001).
This has now been corrected (page 9 para 2).
- Fig3: Reduced DNA fork speed was not rescued with CNOT1.
In Figure 3K the fork speed is appreciably increased with transfected CNOT1, going from 0.46kb/min CNOT1KD) to 0.66 kb/min (complemented with CNOT1). We fail to see the reviewer’s objection.
10) English editing is required.
Grammar has been corrected
